# `IBA`: Towards Irreversible Backdoor Attacks in Federated Learning

**Dung Thuy Nguyen**[1,2*]**, Tuan Nguyen**[2,3]**, Tuan Anh Tran**[4]**, Khoa D Doan**[2,3]**, Kok-Seng Wong**[2,3]

[1] Department of Computer Science, Vanderbilt University, Nashville, TN 37212, USA
[2]VinUni-Illinois Smart Health Center, VinUniversity, Hanoi, Vietnam
[3]College of Engineering & Computer Science, VinUniversity, Hanoi, Vietnam
[4]VinAI Research, Hanoi, Vietnam
`dung.t.nguyen@Vanderbilt.Edu, tuan.nm@vinuni.edu.vn,`
`v.anhtt152@vinai.io, khoa.dd@vinuni.edu.vn, wong.ks@vinuni.edu.vn`

## Abstract

Federated learning (FL) is a distributed learning approach that enables machine learning models to be trained on decentralized data without compromising end devices' personal, potentially sensitive data. However, the distributed nature and uninvestigated data intuitively introduce new security vulnerabilities, including backdoor attacks. In this scenario, an adversary implants backdoor functionality into the global model during training, which can be activated to cause the desired misbehaviors for any input with a specific adversarial pattern. Despite having remarkable success in triggering and distorting model behavior, prior backdoor attacks in FL often hold impractical assumptions, limited imperceptibility, and durability. Specifically, the adversary needs to control a sufficiently large fraction of clients or know the data distribution of other honest clients. In many cases, the trigger inserted is often visually apparent, and the backdoor effect is quickly diluted if the adversary is removed from the training process. To address these limitations, we propose a novel backdoor attack framework in FL, the **Irreversible Backdoor Attack** (`IBA`), that jointly learns the optimal and visually stealthy trigger and then gradually implants the backdoor into a global model. This approach allows the adversary to execute a backdoor attack that can evade both human and machine inspections. Additionally, we enhance the efficiency and durability of the proposed attack by selectively poisoning the model's parameters that are least likely updated by the main task's learning process and constraining the poisoned model update to the vicinity of the global model. Finally, we evaluate the proposed attack framework on several benchmark datasets, including MNIST, CIFAR-10, and Tiny ImageNet, and achieved high success rates while simultaneously bypassing existing backdoor defenses and achieving a more durable backdoor effect compared to other backdoor attacks. Overall, `IBA`[2] offers a more effective, stealthy, and durable approach to backdoor attacks in FL.

## 1 Introduction

Federated Learning (FL) [13] is a novel distributed learning paradigm that enables multiple parties to train machine learning models collaboratively without centralized data sharing. Specifically, each participant trains a local model on their private data, and only the model updates are exchanged with a central server for aggregation. However, FL also introduces new security and privacy challenges

---

[*]Work done while DTN was at VinUniversity.
[2]Code for this paper is published at `https://github.com/sail-research/iba`.

37th Conference on Neural Information Processing Systems (NeurIPS 2023).

that need to be addressed to ensure the confidentiality and integrity of the learning process [33, 24], such as model poisoning, membership inference, data leakage, and Sybil attacks [9, 20].

Backdoor attacks are a specific form of model poisoning in which an attacker inserts a backdoor trigger during the training process that can later be exploited to manipulate the model's behavior [4, 12, 17, 6]. FL is particularly vulnerable to backdoor attacks since each participant trains a local model on their private data, and the central server aggregates the model updates without validating their integrity. An attacker can inject poisoned data or backdoor triggers into the local models of a subset of participants. When the models are aggregated, the backdoor trigger can be activated to compromise the model's behavior [2, 28, 32].

Artificial backdoor attacks are a standard method for backdoors in FL. In an artificial backdoor attack, an apparent trigger, such as a specific pixel value, is injected into the local training data [25, 32]. Although meaningless in the data context, the apparent trigger causes the trained model to associate a signal for the model to optimize during training. When trained on such data, the model will induce misclassifications toward a target class when making predictions on inputs with the presence of the trigger. Existing artificial backdoor attacks require the involvement of a portion of compromised clients and need assistance from scaling-based methods to achieve high attack success rates [32, 24]. This makes these attacks vulnerable to mainstream backdoor defenses [30, 31, 15]. Furthermore, most existing attacks are not durable (i.e., when the adversary is removed from the FL process, the backdoor effect is gradually diluted [34]), especially when the size of compromised clients is small, and they do not regularly participate in the FL training.

In this paper, we exploit the principle of *adversarial attacks* to craft a novel instance-specific backdoor attack, called **Irreversible Backdoor Attack** (IBA), that only requires a small number of compromised clients but achieves more robust attack performance and durability. IBA is a two-phase attack scheme, including *Trigger Generating* and *Backdoor Injection*. During training rounds, the former manipulates the attack model, which is a generative model, so the triggers generated by this attack model can deceive the local model into misbehaving. After that, we formulate the *Backdoor Injection* phase as a constrained optimization problem. Then, we propose a simple but effective recall to keep the attack model updated regularly. This algorithm allows the adversary to achieve high attack success rates even when a small number (i.e., 1) of clients are compromised. In addition, we enhance the durability and effectiveness of the proposed attack by selectively poisoning the model's parameters that are least likely updated by the main task's learning process and constraining the poisoned model update to the vicinity of the global model, respectively. This strategy helps IBA bypass mainstream defenses in FL and achieve extended backdoor durability.

Our **contributions** are summarized below:

- Proposal of a two-phase backdoor injection mechanism, IBA, in FL: The paper introduces IBA, which combines the manipulation of the trigger generator and the poisoning of the local model into a unified two-phase approach. IBA constrains the poisoned model update to be within the vicinity of the global model to improve the practicality and effectiveness of the attack.

- Proposal of selective parameter poisoning in the constrained optimization algorithm: The paper enhances the durability of IBA by selectively poisoning the model's parameters less likely to be updated during the main task's learning process.

- State-of-the-art attack performance and stealthiness: The paper demonstrates that IBA achieves state-of-the-art attack performance and durability and exhibits high stealthiness against existing defense mechanisms. This highlights the robustness and effectiveness of IBA as a practical backdoor attack method.

## 2 Related Work

**Backdoor Attacks in FL.** Backdoor attacks in FL have been extensively studied, revealing vulnerabilities and proposing various attack strategies [16]. Backdoor attackers generally aim to incorporate concealed backdoors into Deep Neural Networks (DNNs) during the training phase. This allows the compromised DNNs to exhibit normal behavior when presented with benign samples. However, their predictions will be deliberately and consistently altered when triggered by specific patterns defined by the attacker. One of the most commonly employed methods for incorporating backdoor

functionality during training is poisoning the training samples [2, 32]. Bagdasaryan et al. [2] also proposed model replacement to implant backdoors into the joint model and demonstrated that even a single-round attack with a small fraction of clients could achieve a 100% Attack Success Rate (ASR). Xie et al.[32] investigated Distributed Backdoor Attacks (DBA), which decompose the global trigger into non-overlapping local triggers. DBA exhibits higher ASR and increased robustness against FL aggregation methods. Additionally, backdoor triggers in machine learning can be designed to be imperceptible in the input space [10, 12, 17, 6] and in the latent space [5], or to possess sophisticated, multi-target payloads [7]; the ground-truth labels of the poisoned samples can also align with the intended target label [26, 22, 35]. These characteristics contribute to the stealthiness of backdoor attacks. Recently, Neurotoxin [34] proposed to enhance backdoor durability by targeting specific coordinates less likely to be updated by benign agents, partially poisoning the model by projecting gradients onto a coordinate-wise constraint. This extends the backdoor's impact and prevents catastrophic forgetting, strengthening the effectiveness of the attack. The recent work that is most aligned with ours is PerDoor [1], which uses the Basic Iterative Method (BIM) to generate the trigger, which relies on gradients of the model's loss function concerning the input data to alter the original inputs into backdoored inputs. However, this method fails to learn underlying patterns and distributions from the training dataset and lacks a sharing and collaborating mechanism among multiple malicious clients, which is very beneficial for the adversary in collaborative attacks.

**Defense Mechanisms against Backdoor Attacks in FL.** The mainstream backdoor-attack defenses in FL aim to reduce the effect on the global model, as it has been shown that detecting backdoors is challenging without breaking the privacy of participants [14]. For instance, Krum [3] aggregates local models by choosing a single local model as the aggregated model with the smallest Euclidean distance to a certain fraction of other models. Norm difference clipping (NDC)[25] examines the norm-difference between the global model sent to and model updates shipped back from selected clients, using a pre-specified threshold to clip model updates that exceed the norm difference. RFA[19] aggregates local models by computing a weighted geometric median using the smoothed Weiszfeld's algorithm. Ozdayi et al. [18] proposed a defense mechanism RLR, that adjusts the learning rate of the aggregation server in FL based on the sign information of agents' updates by carefully considering it per dimension and round. These defense mechanisms offer valuable insights and strategies for countering backdoor attacks in FL settings. Another approach is identifying suspicious clients and removing them from the aggregation process. FoolsGold [8] assumes that benign datasets from different clients differ from each other and assigns low weights to models that are similar to many other models, but it can be vulnerable to sophisticated adversaries submitting poisoned updates without raising suspicion. FLAME [15] uses the HDBSCAN algorithm to detect malicious updates, combines model filtering with poison elimination to detect and remove malicious updates, and is robust against inference attacks.

## 3 Irreversible Backdoor Attack (`IBA`) in FL

**FL Framework.** The training process in the standard FL framework involves the following steps:

*Step 0 (FL Initialization)*: The central server $\mathcal{S}$ initializes the weight of the global model $w$ and sets hyperparameters such as the number of FL rounds and local epochs.

*Step 1 (Local Model Training and Update):* Selected clients $\mathcal{C}_1, \mathcal{C}_2, ..., \mathcal{C}_K$ receive the current global weight $w_0$ from $\mathcal{S}$. Each client $\mathcal{C}_i$ updates its local model parameters $w_i$ using its local dataset $\mathcal{D}_i$. After local training, the clients send their local weights to $\mathcal{S}$ for model aggregation.

*Step 2 (Global Model Aggregation and Update):* $\mathcal{S}$ aggregates the received local weights and computes the aggregation result. The aggregated result is sent back to the clients for the next round of training. $\mathcal{S}$ aggregates the received local model weights using the FedAvg [13] formula:

$$w = \frac{1}{|\mathcal{C}_k|} \sum_{k=1}^{K} |\mathcal{C}_k| \cdot w_k$$

where $|\mathcal{C}_k|$ represents the number of data samples held by client $\mathcal{C}_k$.

**Attacker Threat.** To address the weakness of existing works, i.e., the adversary $\mathcal{A}$ controls a number of compromised clients, we focus on the scenario that $\mathcal{A}$ controls only one client $k$ and it will participate in the training for each $f$ rounds, i.e., fixed-frequency attack [28].

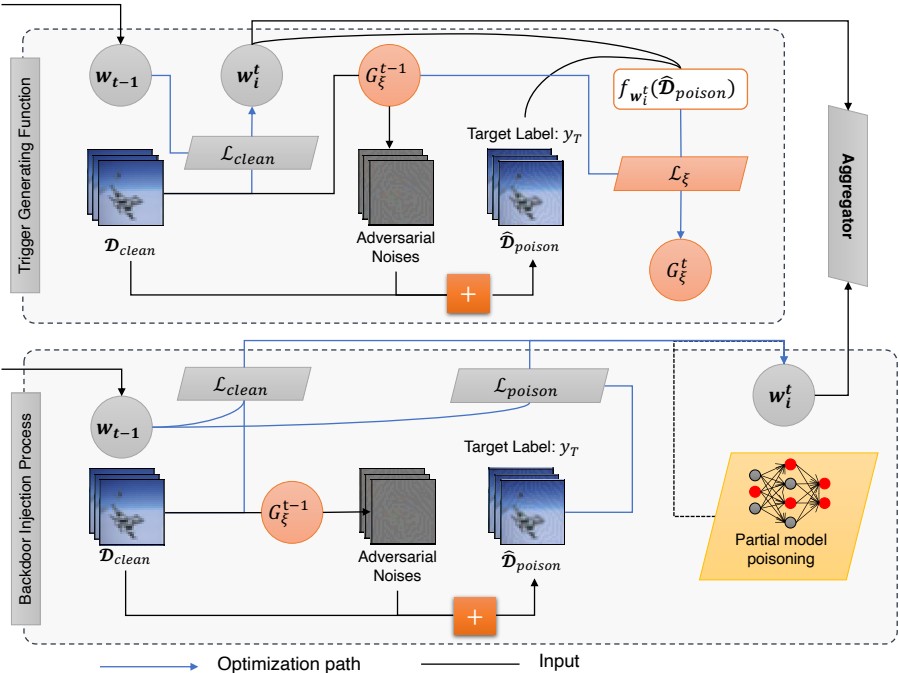

Figure 1: **Irreversible Backdoor Attack Scheme.** IBA includes two training phases: (1) Trigger Generating Function and (2) Backdoor Injection. In the first phase, the trigger generation model $\mathcal{G}_\xi$ is updated to generate adversarial noise. This noise is combined with the original image $x$ to create a backdoor sample to fool the local model $\mathbf{w}_i^t$. In the second phase, the local model $f_{w_k}$ is updated to perform indifferently on clean sample $x$ but distorts its prediction on the backdoor image $T_\xi(x)$ to the target class $y_T$, then this poisoned model is sent to the aggregator. Model poisoning technique is leveraged to enhance the stealthiness and durability of the attack.

*Adversary Capability:* We follow the same assumption used in the existing works on backdoor attacks in FL [2, 15], where adversary $\mathcal{A}$ has total control over the malicious participant(s). However, $\mathcal{A}$ has no control or access to other benign participants' data or local updates. Also, we assume $\mathcal{A}$ thoroughly understands the central server and any potentially deployed defenses but cannot alter the server's configuration parameters and algorithms.

*Adversary Objectives:* The goal of $\mathcal{A}$ is learning a trigger generator $\mathcal{G}_\xi$ such that its outputted triggers can activate the backdoor function to mislead the global model $\mathcal{F}(w)$ while ensuring the behaviors on clean samples are unaffected. This trigger is expected to bypass human and machine inspections. Moreover, $\mathcal{A}$ can bypass configured defenses in FL and prolong the backdoor effect even when $\mathcal{A}$ is removed from the training process. The overall mechanism of our proposed backdoor attack IBA is illustrated in Figure 1.

## 3.1 The Trigger Generating Function

In FL, when clients actively engage in a training round, they have the opportunity to observe and influence the changes in the global model $w$. In particular, a client $k$ receives the global model $w_{t-1}$ and updates it using their local data to produce a new local model, denoted as $w_k$. After local training, the learned surrogate model is aggregated to partially reflect the subsequent global model. In light of this process, we propose an initial phase called the *Trigger Generating*. In this phase, the trigger generation model $\mathcal{G}_\xi$ is updated to generate adversarial noise. This noise is combined with the original image $x$ to create a backdoor sample. This sample remains imperceptibly different from the original, but it effectively misleads the model $\mathcal{G}_\xi$ into making incorrect classifications for the target class. The rationale behind this approach is that manipulating $\mathcal{G}_\xi$ based on this local model allows the adversary $\mathcal{A}$ to optimally adapt the generative function, even in the face of round-by-round variations in the global model. The transformation function is inspired by adversarial examples, in which we model it as a perturbation on the input, as follows:

$$T_\xi(x) = x + \mathcal{G}_\xi(x), ||\mathcal{G}_\xi(x)||_\infty \le \epsilon, \forall x \tag{1}$$

The function $\mathcal{G}_\xi$ takes an input $x$ and generates adversarial noise on the same input space, guaranteeing the backdoor attack's stealthiness.

To achieve this goal, we utilize the local model as the surrogate model to update the attack model $\mathcal{G}_\xi$ using the following objective function:

$$\xi \leftarrow \xi - \eta_\xi \sum_{x \in \mathcal{D}} \mathcal{L}_\xi \left( f_{w_k}(T_\xi(x)), y_T \right), \tag{2}$$

in which $f_w$ is the trained local model and $y_T$ is the targeted label. It's worth noting that the local model $f_w$ submitted to the server will be totally vanilla during this phase.

While considering various norms for bounding the generator noise in Equation 1, such as the $L_2$ norm, it's generally inadvisable for backdoor attacks. This infinity norm guarantees a widespread distribution of the generated trigger across the input image (encompassing all pixels in the trigger); in contrast, the $L_2$ norm can result in localized artifacts within the image (with only select pixels forming the trigger). Employing the $L_2$ norm can make the backdoor attack more susceptible to detection by trigger-synthesis defenses like Neural Cleanse [27]. Thus, the $\|.\|_\infty$ norm is a preferred choice.. We can design such generator function as an auto-encoder or the more complex U-Net architecture [21]. However, we observe no significant performance difference between a simple auto-encoder and U-Net by training the generator function and the classifier with the proposed training algorithm.

In the proposed transformation function, parameter $\epsilon$ controls the visual stealthiness of the triggers. For instance, even on the gray-scale MNIST dataset, if $\epsilon$ is less than 0.01, there is usually no discernible difference between the clean and disturbed images. A greater value for $\epsilon$, i.e., more obvious triggers, simplifies generative model learning. On the other hand, a too-small $\epsilon$ may make learning a generative model challenging since it is heavily dependent on the variance of the surrogate model, which varies across FL training rounds. Therefore, during this phase, we keep the $\epsilon$ sufficiently large, i.e., $\epsilon_0$, until we reach the expected attack success rate.

## 3.2 The Backdoor Injection Process

At the second phase aims to gradually insert the backdoor function into the global model by manipulating the poisoned local model. Our objective is to learn classifier (i.e., local model) $f_{w_k}$ that concurrently performs indifferently on $x$ in comparison to the classifier's clean version but distorts its prediction on the backdoor image $T_\xi(x)$ to the target class $y_T$. The above task can be formulated as the following objective function:

$$\min_w \sum_{x \in \mathcal{D}} \alpha \mathcal{L}_{clean}(f_{w_k}(x), y_x) + \beta \mathcal{L}_{poison}(f_{w_k}(T_\xi(x)), y_T) \tag{3}$$

The classification model $f_w$ in the above problem is trained based on local data and the optimal function $\mathcal{G}_\xi$ learned in the previous phase. When training the classifier, the parameters $\alpha$ and $\beta$ regulate a mixture of the strengths of the loss items from clean and poisoned data, i.e., $\mathcal{L}_{clean}$ and $\mathcal{L}_{poison}$, respectively. If $\alpha$ is more significant than $\beta$, the classifier's performance on clean data rapidly converges to the optimal performance of the vanilla classifier, according to our experiments. Otherwise, the classifier's performance on the backdoor data quickly reaches the optimal value. To compensate for the performance of the local model on both the main task and the backdoor task, we assume $\alpha = 0.5$ and $\beta = 0.5$ in the remaining part of the paper. As previously stated, the value of $\epsilon$ must be assigned cautiously to overcome the unpredictability of the global model and assist the local model in learning the generated triggers more effectively. To achieve this goal, we propose a self-adjusting value for $\epsilon$ from the starting round of this phase, i.e., $t_I$:

$$\epsilon_t = \max(\hat{\epsilon}, \epsilon_0 * (1.0 - \lambda_\xi)^{t-t_I}), \tag{4}$$

where the value of $\epsilon$ is gradually decayed until it reaches objective $\hat{\epsilon}$.

*Attack Model Retraining.* During this phase, we propose frequently updating the attack model $\mathcal{G}_\xi$ to keep it in sync with the state of the global model. The retraining frequency should be chosen (20, 50).

**Partial Model Poisoning.** To enhance the longevity and effectiveness of the attack, we propose to partially poison the model based on selected poisoning space and poisoning dimension. In which space-based poisoning helps the attack bypass the backdoor defenses because it constrains the

poisoned model such that it does not go far from the vicinity of the benign model, while dimension-based poisoning helps to reduce dilution effect from benign clients by shrinking the poisoned neurons to infrequently updated neurons.

*Poisoning-space based.* Under this attack, the submitted local model is partially contaminated by being constrained by the magnitude of its parameters. Specifically, the adversary applies projected gradient descent on the losses for $\mathcal{D} \cup \mathcal{D}'$, where $\mathcal{D}, \mathcal{D}'$ are benign and poisoned datasets, respectively. In other words, it periodically projects the model parameter on the ball centered around the global model of the previous iteration. This enables IBA to circumvent defenses based on norm and distance investigation by preventing the resulting poisoned model from deviating substantially from the original. Mathematically, this method guarantees that $||w_k - w|| \leq \delta$. Note that, this attack strategy can be combined with model replacement [2], where the model parameter is scaled before being sent to the server to cancel the contributions from the other honest clients [28]. The scaled model is calculated by the following:

$$\hat{w}_k \simeq w_k + \sum_{i \in \mathcal{C} \setminus k} \frac{n_i}{n_{C \setminus k}} (w_k - w^*). \tag{5}$$

*Poisoning-dimension based.* Because the attacker cannot participate in every round of training, directly submitting the poisoned model to the server is not effective [31]. To address it, we propose leveraging historical gradient variation to select infrequently updated neurons, which are considered poisoning dimensions for IBA. Then the adversary computes $\text{bottom}-k\%$ coordinates of the gradient vector $g$, obtained by learning the main task on the clean data $\mathcal{D}$. In particular,

$$\beta_t \leftarrow \frac{p-1}{p} * \beta + \frac{1}{p} * \beta_t \quad s.t., \quad \beta_t = bottom_k(g_t), \tag{6}$$

in which $p$ refers to the number of rounds up to round $t$ that adversary $\mathcal{A}$ has been selected to participating in FL training. After that, when finishing calculating gradient $\hat{g}$ of local model $f_w$ using Equation. 3, $\mathcal{A}$ projects gradient onto coordinate-wise constraint on the $\text{top}-k\%$ coordinates of $\hat{g}$. As a result, only $\text{bottom}-k\%$ coordinates are poisoned, which can prolong the backdoor effect since other benign clients rarely update these coordinates. Utilizing historical information aids in determining round-by-round $\text{bottom}-k\%$ coordinates, and it becomes more valuable as rounds advance and local client models converge on the global model.

# 4 Experiments

Our empirical study aims to highlight the effectiveness of the IBA attack against the state-of-the-art (SOTA) FL defenses. We conduct our experiments on real-world datasets and a simulated FL environment. We design three experiment sets: (1) IBA under different attack scenarios, (2) IBA under different FL defenses, and (3) IBA's imperceptibility and durability evaluation. Our results demonstrate that the IBA can achieve significant efficiency, stealthiness, and durability. In particular, IBA attains persistent performance with/without SOTA defenses and maintains higher visual imperceptibility and longer durability compared with existing artificial backdoor attacks [32, 2]. Our implementation is publicly available to reproduce experimental results.

## 4.1 Experimental Setup

**Dataset and Backdoor Tasks.** The IBA method is evaluated on three classification datasets: MNIST, CIFAR-10, and Tiny ImageNet. We simulate heterogeneous data partitioning by sampling $p_k \sim Dir_K(0.5)/Dir_K(0.01)$ for MNIST, CIFAR-10/Tiny ImageNet and allocating a proportion of each class to participating clients. We follow the established protocol outlined in previous works [28, 32], employing stochastic gradient descent (SGD) optimization with $E$ local epochs, a local learning rate of $l_r$, and a batch size of 32. The number of clients selected in each round is $10/200$. The details of the datasets and other parameter configurations are presented in the supplementary material[3].

**Trigger Generation and Backdoored Model Training.** The generation of triggers is accomplished through the utilization of the U-Net architecture [21], incorporating specific parameter configurations.

---

[3]The supplementary material is available at `https://openreview.net/attachment?id=cemEOP8YoC&name=supplementary_material`

Table 1: Effectiveness of IBA under different strategies in fixed-frequency attacks setting ($f = 10$)

| Dataset | Baseline (w/o attack) | | FedAvg | | FedProx | | FedNova | |
|---------|------|------|------|------|------|------|------|------|
| | MA | BA | MA | BA | MA | BA | MA | BA |
| CIFAR-10 | 84.71% | 9.52% | 84.49% | 87.13% | 84.62% | 86.59% | 83.35% | 94.32% |
| MNIST | 99.74% | 9.25% | 98.11% | 99.07% | 97.98% | 99.97% | 98.32% | 92.74% |
| Tiny ImageNet | 65.15% | 0.56% | 64.36% | 94.19% | 65.02% | 92.52% | 64.17% | 93.11% |

The constraint values are set to $\epsilon = 0.3, \hat{\epsilon} = 0.05$; while $\alpha = 0.5$ and $\beta = 0.5$ define additional parameter values. During trigger training, the threshold for backdoor accuracy (BA) is established as $local_{BA} = 0.85$. To facilitate the generation of updates, the learning rate for the update generation model is determined as $\gamma_{\mathcal{A}} = 0.0001$, and for the update classifier model, the learning rate is set to $\eta = 0.01$. Additionally, the performance of IBA is evaluated in conjunction with prominent FL strategies, namely FedAvg [13], FedProx [11], and FedNova [29]. This comprehensive evaluation aims to assess the effectiveness of IBA in the context of different FL methodologies.

**Participating patterns of attackers.** In addition to the *fixed-frequency* case described in the threat model, we further evaluate the efficiency of IBA under a more favorable condition known as *fixed-pool* case (or random sampling), where there is a fixed pool of clients are controlled by $\mathcal{A}$. The result for the fixed-pool case is left in the appendix.

**FL Backdoor Defenses.** We examine six SOTA defense techniques designed to mitigate backdoor attacks in FL: (i) NDC [25], (ii) KRUM and (iii) Multi-KRUM [3], (iv) RFA [19], (v) RLR [18], (vi) FoolsGold [8]. By evaluating the performance of these defense techniques, we aim to provide insights into their effectiveness in countering backdoor attacks in the FL setting.

### 4.2 Experimental Results

**(1) IBA under different attack scenarios.** Firstly, we study the performance of IBA under fixed-frequency attacks with three datasets: MNIST, CIFAR-10, and Tiny ImageNet. The evaluation of main accuracy (MA) and backdoor accuracy (BA) from Table 1 reveals interesting insights. In the absence of the attack, the MAs of the baseline are 99.74%, 84.71%, and 65.15% for MNIST, CIFAR-10, and Tiny ImageNet, respectively, indicating accurate classification on clean samples, while the BA remains low at 0.56% to 9%, indicating no backdoor effect on global model. Under IBA attack with FedAvg aggregator, the corresponding MAs are negligibly affected, while the BAs are notably high, i.e., 99.07% for MNIST dataset. This indicates that IBA can stealthily insert a backdoor function without negatively impacting the model's behavior with benign samples.

In FL, many existing backdoor attacks target vulnerabilities within the FedAvg aggregator since the contribution of each client is considered equally. Therefore, we examined IBA's resilience across different aggregation methods, specifically FedAvg, FedProx, and FedNova, as presented in Table 1. FedProx, with its proximal term, enhances model accuracy and is beneficial in scenarios with non-IID data distributions. Besides, FedNova, designed to address objective inconsistency in FL, ensures consistent and accurate model updates even with varying update frequencies from clients, making it suitable for scenarios with heterogeneous data.

The results demonstrate the consistent efficacy of IBA with three datasets and three FL aggregators. In particular, IBA persistently obtained significant BAs, with the lowest BA being 87.13% and the highest being 99.77%, demonstrating its efficacy and potential challenges in diverse FL settings.

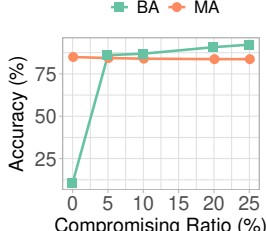

**Fixed-pool attacks.** We study the performance of IBA under fixed-pool attack setting (percentage of attackers in the overall clients varies from 5% to 20%). The results are shown in Figure 2, demonstrating that a higher compromising ratio leads to a better backdoor accuracy or more substantial backdoor effect.

Figure 2: IBA under fixed-pool attacks with different compromising ratios.

**(2) IBA under different FL defenses.** We study the effectiveness of standalone IBA and hybrid attacks (i.e., IBA and partial model poisoning) against the aforementioned defense techniques. As demonstrated in

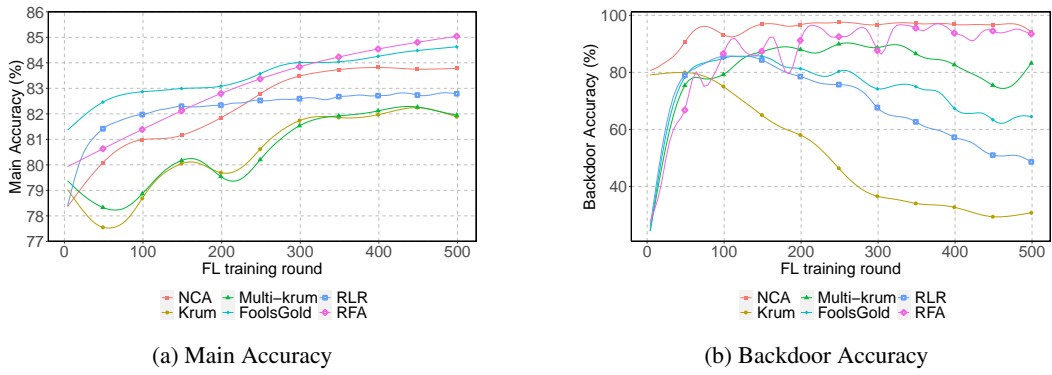

(a) Main Accuracy                    (b) Backdoor Accuracy

Figure 3: Effectiveness of stand-alone IBA against mainstream defenses for FL on CIFAR-10 dataset.

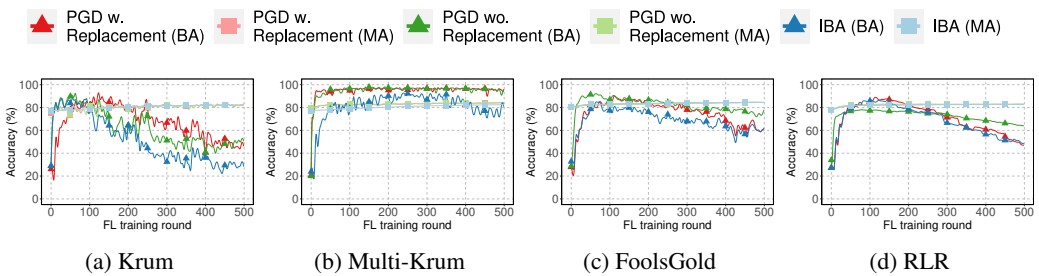

(a) Krum          (b) Multi-Krum          (c) FoolsGold          (d) RLR

Figure 4: The improved stealthiness of IBA when combined with partial model poisoning attacks under various defenses for the CIFAR-10 dataset. The attack is conducted from round 0 with fixed-frequency attacks of 10.

Figure 3, the stand-alone IBA can circumvent NCA, Multi-Krum, and FoolsGold and attain a high backdoor accuracy (i.e., 95.11%, 79.5% and 76.46% for CIFAR-10 dataset, respectively). Krum is the most challenging defense since it replaces the global model with only the most representative local model (i.e., the one with the shortest distance to its neighbors) in each round. As a result, the global model experiences substantial fluctuation over rounds and does not fully capture the knowledge of all participants, i.e., a significant decline in primary task accuracy in Figure 3. This mechanism hampers

Table 2: Robustness of IBA and the combination of IBA and partial model poisoning under mainstream backdoor defenses in FL

| Dataset | Defense | IBA (Standalone) | | IBA + PGD | | IBA + PGD + MR | |
|---|---|---|---|---|---|---|---|
| | | MA | BA | MA | BA | MA | BA |
| CIFAR-10 | RFA | 81.97% | 98.03% | 80.71% | 96.7% | 81.25% | 98.17% |
| | NCA | 84.64% | 95.11% | 84.77% | 98.73% | 84.21% | **99.12%** |
| | Krum | 81.65% | 27.07% | 81.59% | 50.33% | 81.44% | 65.98% |
| | Multi-krum | 82.40% | 79.5% | 83.78% | **97.16%** | 83.25% | 95.02% |
| | RLR | 83.91% | 63.94% | 82.59% | 65.14% | 82.85% | 47.07% |
| | FoolsGold | **84.90%** | 76.46% | **85.02%** | 76.78% | 82.90% | 84.36% |
| | No-defense | 84.28% | 85.63% | 84.35% | 83.80% | **83.81%** | 97.79% |
| MNIST | RFA | 98.04% | **100.0%** | 98.89% | 99.64% | 98.04% | **100.0%** |
| | NCA | 98.66% | **100.0%** | 98.97% | **99.69%** | 98.62% | 100.0% |
| | Krum | 97.60% | 14.13% | 97.82% | 87.48% | 97.44% | 17.29% |
| | Multi-krum | 97.98% | 81.27% | 98.97% | 98.82% | 99.01% | 80.18% |
| | RLR | 90.91% | 12.60% | 87.48% | 18.74% | 90.03% | 29.97% |
| | FoolsGold | 98.87% | 80.08% | **99.10%** | 96.43% | **99.05%** | 99.83% |
| | No-defense | **99.98%** | 98.03% | 98.81% | 99.73% | 98.84% | 99.97% |

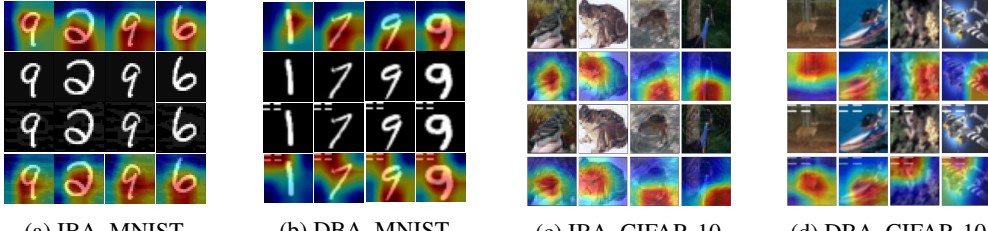

| (a) IBA, MNIST | (b) DBA, MNIST | (c) IBA, CIFAR-10 | (d) DBA, CIFAR-10 |

Figure 5: Performance under GradCam heat maps of two attack schemes: IBA and Patched-BA with MNIST and CIFAR-10 datasets. The first two rows in each image represent the benign samples, and the last two rows represent the backdoored ones.

the attack model's learning process since it is closely associated with global model variance. Figure 4 illustrates the improved performance of IBA against challenging defenses when combined with partial model poisoning, i.e., PGD-based. As observed, the IBA + PGD (w./wo.) model replacement can help improve the backdoor accuracy from 10% to 20%. Significantly, the improved IBA can achieve almost 100% of backdoor accuracy for Multi-Krum defense. The more detailed result can be seen in Table 2. The proposed IBA can bypass most existing defenses and achieve significant backdoor accuracy. In most cases, the combination of IBA + PGD/ IBA + PGD + MR helps improve the backdoor accuracy from 10% to 20%. While Krum and RLR defend the proposed attack to some extent, i.e., we observe the drop in the BAs of IBA, we argue that Krum and RLR tend to reject previously unseen information from both adversary and honest clients, which results in low MAs (a drop of 3–10%). Specifically, the proposed IBA causes RLR to compensate for the backdoor defense's effectiveness with the main accuracy drop in the MNIST dataset by unnecessarily reversing the learning rate of specific dimensions, i.e., the BAs are around 90% (less than other defenses 10%). Therefore, these two defenses raise great concern about its practicability under this scenario.

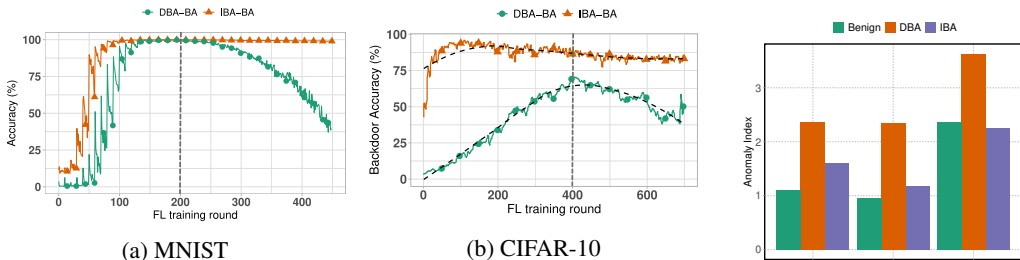

| (a) MNIST | (b) CIFAR-10 |

Figure 6: Durability comparison of IBA and DBA with MNIST and CIFAR-10 datasets. The adversary is removed from rounds 200 and 400 training, respectively.

Figure 7: Anomaly Indexes of IBA, DBA backdoor attacks in FL.

Table 3: The decay rate of DBA and IBA after removing the backdoor attack (i.e., from the stopping round). The column (%) indicates the current backdoor effect as a percentage of its peak value at the stopping round.

| Dataset | Method | Stopping round | | 50 rounds later | | 100 rounds later | | 150 rounds later | | 200 rounds later | | 250 rounds later | |
|---|---|---|---|---|---|---|---|---|---|---|---|---|---|
| | | BA | (%) | BA | % | BA | (%) | BA | (%) | BA | (%) | BA | (%) |
| MNIST | IBA | 99.94 | 100.00 | 99.84 | 99.90 | 99.52 | 99.57 | 99.46 | 99.52 | 99.18 | 99.24 | 99.08 | 99.14 |
| | DBA | 99.70 | 100.00 | 98.44 | 98.74 | 91.44 | 91.71 | 79.88 | 80.12 | 65.04 | 65.24 | 38.64 | 38.76 |
| CIFAR-10 | IBA | 87.51 | 100.00 | 86.22 | 98.53 | 85.32 | 97.50 | 84.89 | 97.00 | 83.19 | 95.06 | 83.16 | 95.03 |
| | DBA | 69.63 | 100.00 | 65.55 | 94.14 | 63.59 | 91.32 | 55.39 | 79.55 | 55.07 | 71.91 | 42.55 | 61.11 |

**(3) IBA's imperceptibility and durability evaluation.** Besides being imperceptible to the backdoor defenses, IBA is expected to be visually imperceptible and deliver prolonged backdoor durability. The visual imperceptibility of IBA is illustrated in Figure 5. We evaluate the behavior of backdoor-injected models against GradCam [23], a widely known visualization tool and helpful in understanding the neural network's behavior. Figure 5 shows that the visualization heat maps created by IBA attacks are nearly identical across the clean and backdoor images. From Figures 5, we can observe that the

backdoored images have negligible deviations from the Grad-CAM behaviors on the clean images for both black-and-white and colored images. On the other hand, backdoored images using patched triggers generate a considerable difference in visualization heat maps (DBA [32]). Because IBA is based on input perturbation, the image is formed by adding very little perturbation to the clean image; consequently, the difference in the latent space where the heat maps are generated is likewise minimal. Aside from that, the patch backdoor (such as the pattern in the upper-left corner) is apparent, resulting in a significant distinction between the backdoor and clean versions. We also evaluate the backdoor generated by IBA and another attack scheme in FL, i.e., DBA with **Neural Cleanse** [27], a widely-used backdoor model mitigation method based on the pattern optimization approach. Neural Cleanse assumes that the backdoor trigger is patch-based, which makes it suitable for evaluating the proposed method. For each image label, Neural Cleanse identifies if there is a patch pattern that produces a misclassification result for that target label. If any class label yields a significantly smaller pattern, Neural Cleanse considers it a sign of a potential backdoor. Neural Cleanse quantifies such deviations from the optimal patch of each class label by using the Anomaly Index metric. If the Anomaly Index is less than a threshold of 2 for a class, Neural Cleanse considers that there is a backdoor with this class as the target label. Since IBA does not generate the patch-based trigger pattern, it can efficiently bypass this defense by having *Anomaly Index* much lower than DBA (Figure 7), which indicates the superior stealthiness of backdoor attacks with adversarial triggers compared to the patch-based.

As previously stated, strong attacks should be successful even if the attacker only participates a few times. As a result, we construct a scenario to assess the endurance of IBA compared to a chosen baseline: DBA. We employ the same configuration as the original work DBA but limit each of the four malicious clients' participation frequency to 10 without utilizing any scaling strategy. Regarding IBA, we apply the dimension-based model poisoning mentioned above and use one malicious client with the same frequency. As observed in Figure. 6a, after $\mathcal{A}$ is removed from the training (i.e., at round 200), the BA of DBA reduces gradually while the accuracy drop of IBA is almost negligible. As demonstrated in Figure 6b and Table 3, IBA delivers superior backdoor efficiency and durability within the same attacking duration. After round 400, the BA of DBA drops quickly compared to IBA. We can see the most significant difference towards the end of the comparison duration, i.e., at round 700, the BAs of IBA and DBA are $\sim 85\%$ and $\sim 45\%$, respectively. After 250 rounds of elimination, the BA of DBA retains 38.76% of its peak value, but the comparable value of IBA is 99.14%. Furthermore, while the BA can preserve more than 60% of its peak value with the CIFAR-10 dataset, the highest accuracy it can attain is significantly lower than IBA, i.e., less than 20%, which is consistent with the prior conclusion of the original when the scaling factor is 1. In conclusion, IBA can achieve better durability and imperceptibility than SOTA attacks.

## 5    Conclusion

In conclusion, we have presented the IBA, a novel backdoor attack framework for FL that overcomes the limitations of existing approaches. Our framework combines a learnable trigger-generating function with a gradual implantation process, resulting in an attack that can evade detection and achieve high success rates. The selective poisoning of model parameters and the constraint on poisoned updates enhance the attack's efficiency and durability. Our work demonstrates promising results and suggests several directions for future research. Robust defense mechanisms against FL backdoor attacks must be developed, considering the unique challenges of the distributed and decentralized learning process. Generalizing our approach to diverse FL settings, such as heterogeneous or non-i.i.d. data distributions, would provide a comprehensive understanding of the attack's effectiveness. Exploring alternative triggers and their impact on stealthiness and success rates could yield valuable insights. Evaluating the framework on larger-scale FL systems and real-world scenarios will assess its scalability and practicality. Advancing adversarial defense strategies, including proactive detection and mitigation techniques, will contribute to more secure and trustworthy FL systems. These future research directions will contribute to strengthening the security of FL against backdoor attacks.

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
