# Supplementary Material
# `IBA`: Towards Irreversible Backdoor Attacks in Federated Learning

**Dung Thuy Nguyen**[1,2*]**, Tuan Nguyen**[2,3]**, Tuan Anh Tran**[4]**, Khoa D Doan**[2,3]**, Kok-Seng Wong**[2,3]

[1] Vanderbilt University, Nashville, TN 37235
[2]VinUni-Illinois Smart Health Center, VinUniversity, Hanoi, Vietnam
[3]College of Engineering & Computer Science, VinUniversity, Hanoi, Vietnam
[4]VinAI Research, Hanoi, Vietnam
dung.t.nguyen@Vanderbilt.Edu, tuan.nm@vinuni.edu.vn,
v.anhtt152@vinai.io, khoa.dd@vinuni.edu.vn, wong.ks@vinuni.edu.vn

This document serves as an extended exploration of our research, offering an overview of our methods and results. In Appendix A, we provide a detailed account of the training process, datasets, model structures, and the specific configurations used to create the optimal triggers employed in the proposed Irreversible Backdoor Attack (`IBA`) for Federated Learning (FL). Moving beyond the technical details, in Appendix B, we present a wealth of experimental results showcasing the effectiveness of `IBA` in bypassing common FL defenses. Appendix C offers a closer look at ablation studies, which reinforce `IBA`'s long-lasting impact and its ability to withstand diverse scenarios even after the removal of the backdoor, distinguishing it from other state-of-the-art attacks. In Appendix D, we delve into the limitations of `IBA`, shedding light on potential areas of improvement. Finally, in Appendix E, we engage in a discussion about the broader implications of `IBA`.

## A    Training Details and Experimental Settings for `IBA` in FL

We conduct all the experiments using PyTorch version 2.0 [13] and refer to the frameworks provided by [17] and [18] for implementing backdoor attacks in FL.

### A.1    Dataset

We uses three datasets that are widely used for image classifications task with non-i.i.d. data distribution, i.e., MNIST, CIFAR-10 and Tiny ImageNet. The data description and parameter setups are summarized in Table 1.

- **MNIST** [3]: dataset of 60,000 small square $28 \times 28$ pixel grayscale images of handwritten single digits between 0 and 9.

- **CIFAR-10** [6]: is an established computer-vision dataset used for object recognition. It is a subset of the 80 million tiny images dataset and consists of 60,000 $32 \times 32$ color images containing one of 10 object classes, with 6000 images per class.

- **Tiny ImageNet** [7]: Tiny ImageNet contains 110000 images of 200 classes (550 for each class) downsized to 64×64 colored images. Each class has 500 training images and 50 test images.

---

*Work done while DTN was at VinUniversity.

37th Conference on Neural Information Processing Systems (NeurIPS 2023).

Table 1: Dataset description and parameters under `IBA` backdoor attack.

|  | MNIST | CIFAR-10 | Tiny ImageNet |
|---|---|---|---|
| # Samples | 60,000 | 60,000 | 110,000 |
| Total Clients | N = 200 | N = 200 | N = 200 |
| Clients/Rounds | K = 10 | K = 10 | K = 10 |
| Model | LeNet [8] | VGG-9 [15] | Resnet-18 [5] |

## A.2 Experimental setup.

We implemented the proposed `IBA` attack on a single machine with the following hardware configuration: an 64-core Intel Xeon CPU with a GeForce RTX 3090 GPU to accelerate training. Our codebase primarily relied on the *PyTorch 2.0* framework for deep learning tasks. Additionally, we utilized utilities from *Scikit-learn, Pandas*, and *Matplotlib* to support various functionalities in our experiments. Our simulated FL environment follows [1, 17] where for each FL round, the data center selects a subset of available clients and broadcasts the current model to the selected clients. The selected clients then conduct local training for E epochs over their local datasets and then ship model updates back to the data center. The data center then conducts model aggregation (e.g. weighted averaging in FedAvg). The FL setups in our experiment are inspired by [17, 1, 18], the number of total clients, number of clients participates per FL round, and the specific choices of E for various datasets in our experiment are summarized in Table 2. For the task with MNIST, we retrain a LeNet model from scratch. Regarding tasks with CIFAR-10/Tiny ImageNet datasets, our FL process starts from a VGG-9 model with 77.68% test accuracy, a Resnet18 model with 63.35% accuracy top-1 accuracy, respectively.

Table 2: Dataset specifications and parameters

| Dataset | Classes | Features | Model | Benign $l_r/E$ | Poison $l_r/E$ | Batch size/ Test batch size |
|---|---|---|---|---|---|---|
| MNIST | 10 | 784 | LeNet | 0.01 / 2 | 0.01 / 2 | 32/256 |
| CIFAR-10 | 10 | 1024 | VGG-9 | 0.02 / 2 | 0.02 / 2 | 32/256 |
| Tiny ImageNet | 200 | 4096 | Resnet-18 | 0.001 / 2 | 0.001 / 2 | 256/256 |

Followed by [17], we simulate heterogeneous data partitioning by sampling $\mathbf{p}_k \sim Dir_K(\varphi)$ and allocating a $\mathbf{p}_{k,i}$ proportion of $\mathcal{D}^k$ of class $k$ to local user $i$. Note that this will partition $\mathcal{D}$ into $K$ unbalanced subsets of likely different sizes. Under this strategy, fraction of $\varphi$ samples of each class is assigned to group clients associated with this class, and the remaining is randomly distributed to others. As a result, non-IID degree $\varphi$ is zero means the data is distributed completely IID (homogeneity), likewise, the data distribution is absolutely non-IID when $\varphi$ equals one. The value of $\varphi$ is set to be 0.5 in all our experiments, which is align with prior works [17, 18].

## A.3 Hyper-parameters used with IBA.

**Backdoor learning's harmony-controlling parameters** $\alpha, \beta$**.** When training the classifier, the parameters $\alpha$ and $\beta$ regulate the mixing strengths of the loss signals from the clean and backdoor data. In our studies, we find that if $\alpha$ is greater than $\beta$, the classifier's performance on clean data quickly converges to the vanilla classifier's optimum. When $\beta$ is more significant than $\alpha$, the classifier's performance on the backdoor data rapidly achieves the perfect value. However, if the trigger generator $\mathcal{G}$ is properly trained, the backdoor classifier converges to the same optimal performances on both clean and backdoor data. For these reasons, in the remaining part of the paper, we assume $\alpha = 0.5$ and $\beta = 0.5$ for the MNIST and CIFAR-10 datasets. Tiny ImageNet, on the other hand, is a difficult dataset to backdoor, especially when the data distribution is highly non-iid, because the local model is prone to over-fitting. To circumvent this, we used $\alpha = 0.7$ and $\beta = 0.3$ in our studies with this dataset.

**Learning rate** $\eta_{\hat{\xi}}$ **of attack model** $\mathcal{G}$**.** This setting governs the trigger generator's speed of learning. We observed that t $\eta_{\hat{\xi}} = 0.0001, 0.0005$ are appropriate for MNIST/CIFAR-10 and Tiny

ImageNet, respectively, by empirical experiments. The learning rate $\eta_{\hat{\xi}}$ is suggested to be in range $(0.0001, 0.001)$.

**Number of local training epoch $\varepsilon$ for $\mathcal{G}$.** In both phases, the trigger generator $\mathcal{G}$ is updated for predefined $\varepsilon$ iterations. This parameter should be adjusted accordingly to the convergence speed of the targeted dataset. For instance, with the MNIST dataset, we keep the value for $\varepsilon$ at 3 for both phases. Regarding CIFAR-10 and Tiny ImageNet, $\varepsilon$ is set to 10 and 7, respectively. Since the training of generator $\mathcal{G}$ is completely exclusive at the client side and is performed after local model training is finished, it can be conducted even when the adversary $\mathcal{A}$ does not participate in the training. Furthermore, this factor can be manually adjusted to help the backdoor be more effective, i.e., the $\varepsilon$ can be increased to 10 for Tiny ImageNet at the retraining steps of the attack model.

**Expected backdoor accuracy $\lambda$.** This value serves as a threshold for transitioning from phase 1 (*attack model's warm-up*) to phase 2 (*backdoor insertion*). When the adversary engages in the training, the local backdoor accuracy is calculated, and the adversary enters the second phase if it is larger than the $\lambda$ threshold. In all experiments, we used the value $\lambda = 0.85$. The smaller the $\lambda$, the faster the local model will be poisoned. However, a low threshold may make it difficult for the local model to learn both the backdoor and the main job, resulting in a considerable deviation from the local model to the benign ones.

**Decay rate $\lambda_{\xi}$.** $\lambda_{\xi}$ is the decay rate controling how fast the reduction by round is. The larger $\lambda_{\xi}$ is, the faster the backdoor images become stealthy and the harder it is for the generative model to learn to perform well, i.e., it may overfit with the local data. Therefore, via empirical study, we fix the value of $\lambda_{\xi}$ to be 0.001 to balance out the two factors mentioned above. Our suggestion is that $(1 - \lambda_{\xi})$ should be set in the range $[1 \times (1 - \gamma); 3 \times (1 - \gamma)]$, where $\gamma$ is the decay factor of learning rate, i.e., in our inherited setup, $\gamma$ is set to be 0.998.

**Norm-bound for PGD attack.** The norm-bound for PGD attack should be selected based on the observation of the L2-norm variation of the global model by round. Following the work [1] and [17], the norm bound is set to be 2 for the CIFAR-10/Tiny ImageNet dataset and 1.5 for the MNIST dataset.

### A.4 Hyper-parameters used within the defense mechanisms.

With the implementations of Krum/Multi-Krum [2], FoolsGold [4], RFA [14], RLR [12], NDC/NCA [16], the majority of hyper-parameters are inherited with minor modifications. There are several parameters that are adjusted to make experiment settings more appropriate. Other hyperparameters without mention are setup as in the original works.

**Norm-clipping threshold in NDC/NCA [16].** In our experiments, we set the norm difference threshold at 2, which is followed the previous work [17].

**Robust threshold $\theta$ of RLR [12].** The value of $\theta$ in the RLR method is specified to be any value between $[K \cdot F + 1, K - K \cdot F]$, where K is the number of participants each round and F is the proportion of malicious clients, according to the authors. Because there is limited to one malicious client each round, the value of $\theta$ is set to 1 throughout the experiments with all datasets.

**Estimated number of Byzantine clients $F$ in Krum/Multi-Krum [2].** Since the experiments are conducted under fixed-frequency backdoor attacks with maximum of one malicious client, so $F$ is set to be 1.

## B Additional experiments IBA

### B.1 Stealthiness of IBA under defenses

The evaluation of the backdoor attack in FL was conducted using three datasets: MNIST, CIFAR-10, and Tiny ImageNet. The experimental results, including the performance of the IBA under various backdoor defenses, are summarized in Table 3. The fluctuation of MAs and BAs by rounds of IBA under these defenses is illustrated in Figure 1, 2, 3. In addition, the performance of IBA combined with different model poisoning techniques with MNIST and Tiny ImageNet datasets can be observed in Figure 4, 5.

Table 3: Robustness of IBA under mainstream backdoor defenses in FL

| Dataset | Defense | IBA (Standalone) | | IBA + PGD | | IBA + PGD + MR | |
|---|---|---|---|---|---|---|---|
| | | MA | BA | MA | BA | MA | BA |
| CIFAR-10 | RFA | 81.97% | 98.03% | 80.71% | 96.7% | 81.25% | 98.17% |
| | NCA | 84.64% | 95.11% | 84.77% | 98.73% | 84.21% | **99.12%** |
| | Krum | 81.65% | 27.07% | 81.59% | 50.33% | 81.44% | 65.98% |
| | Multi-krum | 82.40% | 79.5% | 83.78% | **97.16%** | 83.25% | 95.02% |
| | RLR | 83.91% | 63.94% | 82.59% | 65.14% | 82.85% | 47.07% |
| | FoolsGold | **84.90%** | 76.46% | **85.02%** | 76.78% | 82.90% | 84.36% |
| | No-defense | 84.28% | 85.63% | 84.35% | 83.80% | **83.81%** | 97.79% |
| MNIST | RFA | 98.04% | **100.0%** | 98.89% | 99.64% | 98.04% | **100.0%** |
| | NCA | 98.66% | **100.0%** | 98.97% | **99.69%** | 98.62% | 100.0% |
| | Krum | 97.60% | 14.13% | 97.82% | 87.48% | 97.44% | 17.29% |
| | Multi-krum | 97.98% | 81.27% | 99.01% | 80.18% | 98.97% | 98.82% |
| | RLR | 90.91% | 12.60% | 87.48% | 18.74% | 90.03% | 29.97% |
| | FoolsGold | 98.87% | 80.08% | **99.10%** | 96.43% | **99.05%** | 99.83% |
| | No-defense | **99.98%** | 98.03% | 98.81% | 99.73% | 98.84% | 99.97% |
| Tiny ImageNet | RFA | 65.09% | 90.70% | 65.07% | 88.65% | 65.05% | 92.74% |
| | NCA | 65.00% | 91.97% | 64.94% | **93.67%** | 64.97% | 93.33% |
| | Krum | 63.53% | 86.57% | 63.3% | 85.67% | 63.65% | 86.36% |
| | Multi-krum | 65.16% | 87.03% | 65.08% | 87.57% | 65.14% | 85.16% |
| | RLR | 63.28% | 89.67% | 63.29% | 89.09% | 63.3% | 86.52% |
| | FoolsGold | 64.84% | 91.39% | 64.8% | 89.9% | 64.73% | 85.72% |
| | No-defense | **65.21%** | **94.51%** | **65.22%** | 85.66% | **65.21%** | 93.53% |

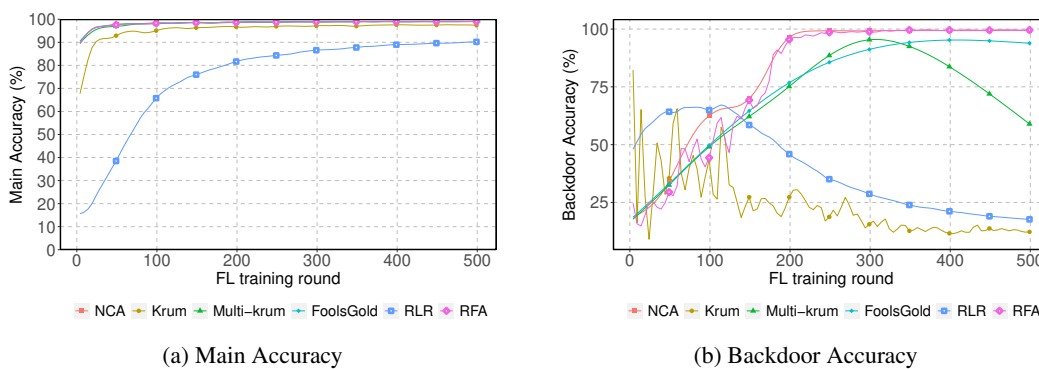

(a) Main Accuracy

(b) Backdoor Accuracy

Figure 1: Effectiveness of stand-alone IBA against mainstream defenses for FL on MNIST.

Based on the observations for MNIST, CIFAR-10, and Tiny ImageNet datasets, we can summarize the effectiveness of IBA under different FL defenses as follows:

- Stand-alone IBA: The stand-alone IBA attack demonstrates its capability to circumvent certain defenses and achieve high backdoor accuracy. Specifically, it can bypass NCA, Multi-Krum, and FoolsGold defenses, resulting in backdoor accuracies ranging from approximately 60% to nearly 100%. However, Krum defense poses the most significant challenge for the stand-alone IBA. Krum replaces the global model with only the most representative local model in each round, leading to substantial fluctuations in the global model and a decline in primary task accuracy. This mechanism hampers the learning process of the attack model as it heavily relies on the stability of the global model.

- Hybrid Attacks: The combination of IBA with partial model poisoning, specifically PGD-based techniques, enhances the performance of IBA against challenging defenses. The inclusion of PGD-based model replacement improves the backdoor accuracy, ranging from 10% to 20%. Notably, the improved IBA achieves almost 100% backdoor accuracy against

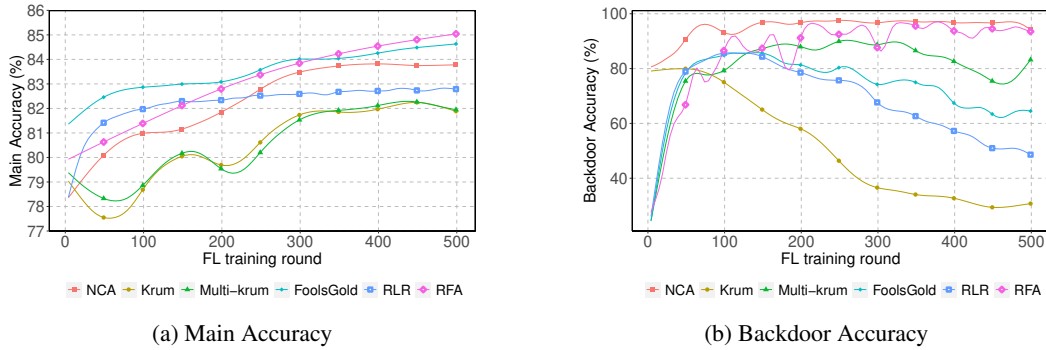

(a) Main Accuracy

(b) Backdoor Accuracy

Figure 2: Effectiveness of stand-alone `IBA` against mainstream defenses for FL on CIFAR-10 dataset.

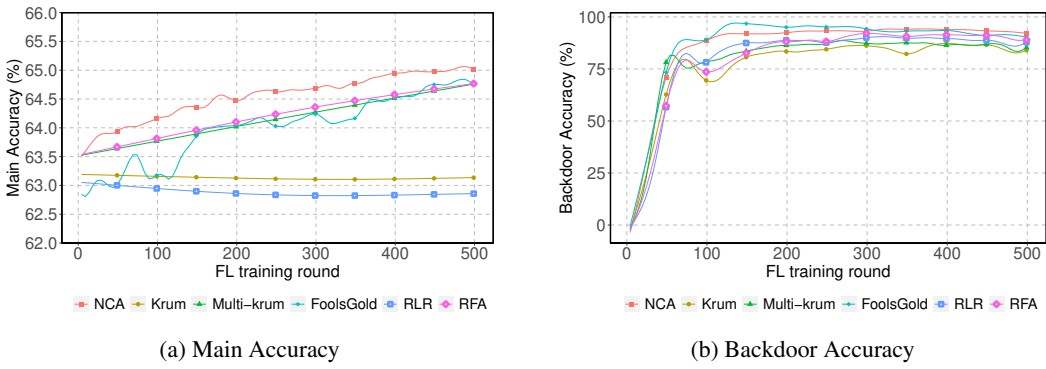

(a) Main Accuracy

(b) Backdoor Accuracy

Figure 3: Effectiveness of stand-alone `IBA` against mainstream defenses for FL on Tiny ImageNet.

the Multi-Krum defense. This demonstrates the effectiveness of the hybrid approach in enhancing the bypassing capabilities of `IBA`.

- Defense Trade-offs: While the proposed `IBA` can bypass most existing defenses and achieve significant backdoor accuracy, certain defenses such as RLR and Krum impose trade-offs between defensive efficiency and primary task performance. For example, under the RLR defense, there is a 10% drop in accuracy on the MNIST dataset, raising concerns about its feasibility in this scenario. These trade-offs highlight the challenges faced when balancing defense effectiveness and maintaining acceptable primary task performance.

In summary, the evaluation of IBA under different FL defenses reveals the circumvention capabilities of stand-alone IBA against certain defenses and the improved performance achieved through hybrid attacks. However, the trade-offs between defense efficiency and primary task performance should be carefully considered when selecting and deploying defense mechanisms in practical scenarios.

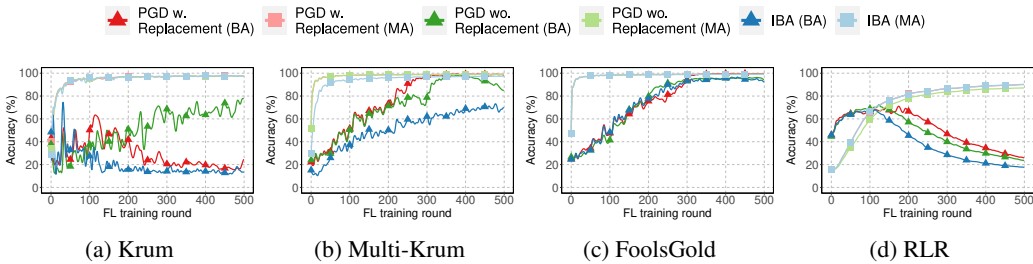

(a) Krum

(b) Multi-Krum

(c) FoolsGold

(d) RLR

Figure 4: The improved stealthiness of `IBA` when combined with partial model poisoning attacks under various defenses for MNIST.

**Observations with Tiny ImageNet.** As shown in Table 3, IBA attains consistent performance on the Tiny ImageNet. Even against difficult defenses like RLR and Krum, IBA can obtain BAs of approximately 85%. The fundamental reason is that the Resnet-18 model is significantly larger

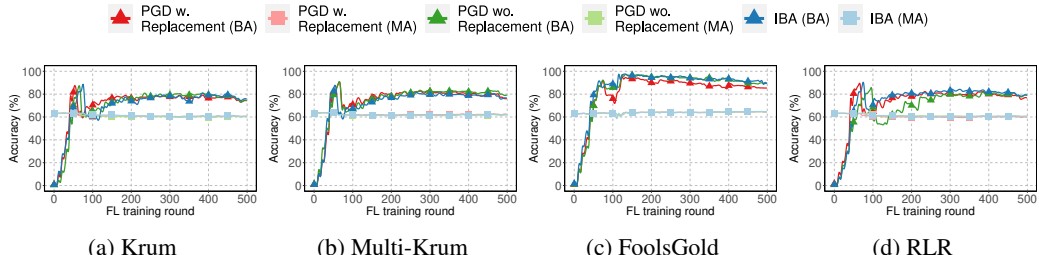

Figure 5: The improved stealthiness of `IBA` when combined with partial model poisoning attacks under various defenses for Tiny ImageNet dataset.

than the LeNet and VGG-9 models used in the MNIST/CIFAR-10 dataset, therefore when the local model is poisoned, the difference between poisoned neurons and benign ones may come unpredictable. Furthermore, Tiny ImageNet contains 200 classes, and data distribution amongst clients is substantially divergent, resulting in a large variance between local models. As a result, the performance of these defenses may be negatively impacted. In addition, the main accuracy of the classification task with Tiny ImageNet does not improve significantly under such FL settings, which is analogous with the observation in [18], which is also a benefit for IBA to learn the optimal trigger generation by rounds.

## B.2 Stability of `IBA` under different settings

### Results under a different number of clients.

We adopt standard settings with 10 clients, as seen in previous works [1, 18]. However, our study also explores scenarios with different client counts, specifically 5 and 20 clients. Remarkably, `IBA` consistently performs well across these variations, as evident in Figure 6. In this experiment, we extended our evaluation with FLAME [11] defense, which . . . . The results show that even stand-alone IBA can bypass this defense, i.e., IBA can achieve BAs of 99% with the task of MNIST for both cases of K (K is the number of clients selected in each round): 5 and 20.

**Results under different settings of data distribution.** In FL, data distribution across parties is typically non-i.i.d. Following established protocols [1, 17, 18], we generate diverse data distributions using Dirichlet distributions [10], i.e., $p_k \sim Dir_K(\varphi)$ with different hyperparameter $\varphi$. Then, we allocate a $p_{k,i}$ proportion of class $k$ to local client $i$. The result is the whole dataset will be partitioned into $K$ unbalanced subsets of likely different sizes. As a result, a non-IID degree $\varphi$ is zero means the data is distributed completely IID (homogeneity), likewise, the data distribution is non-IID when $\varphi$ equals one. Without any clarification, we set up $p_k \sim Dir_K(0.5)$ for MNIST, CIFAR-10, and $p_k \sim Dir_K(0.01)$ for Tiny Imagenet.

We explore various values of $\varphi \in (0.2, 0.5, 0.7)$ within the Dirichlet distribution, simulating transitions from non-i.i.d. to i.i.d. data distributions for image datasets. Our evaluation encompasses two distinct datasets, highlighting the stability of `IBA` across diverse distributions. This underscores the practicality and resilience of `IBA` when employed in conventional FL settings. Table 4 below presents the results of `IBA` under different $\varphi$ values in the Dirichlet distribution.

Table 4: Main Accuracy (MA) and Backdoor Accuracy (BA) of `IBA` under different $\alpha$ values in the Dirichlet distribution.

| Dataset | $\varphi = 0.2$ | | $\varphi = 0.5$ | | $\varphi = 0.7$ | |
|---|---|---|---|---|---|---|
| | MA | BA | MA | BA | MA | BA |
| MNIST | 99.01% | 98.89% | 99.98% | 98.03% | 98.99% | 98.81% |
| CIFAR-10 | 82.73% | 83.26% | 84.28% | 85.63% | 84.53% | 88.93% |

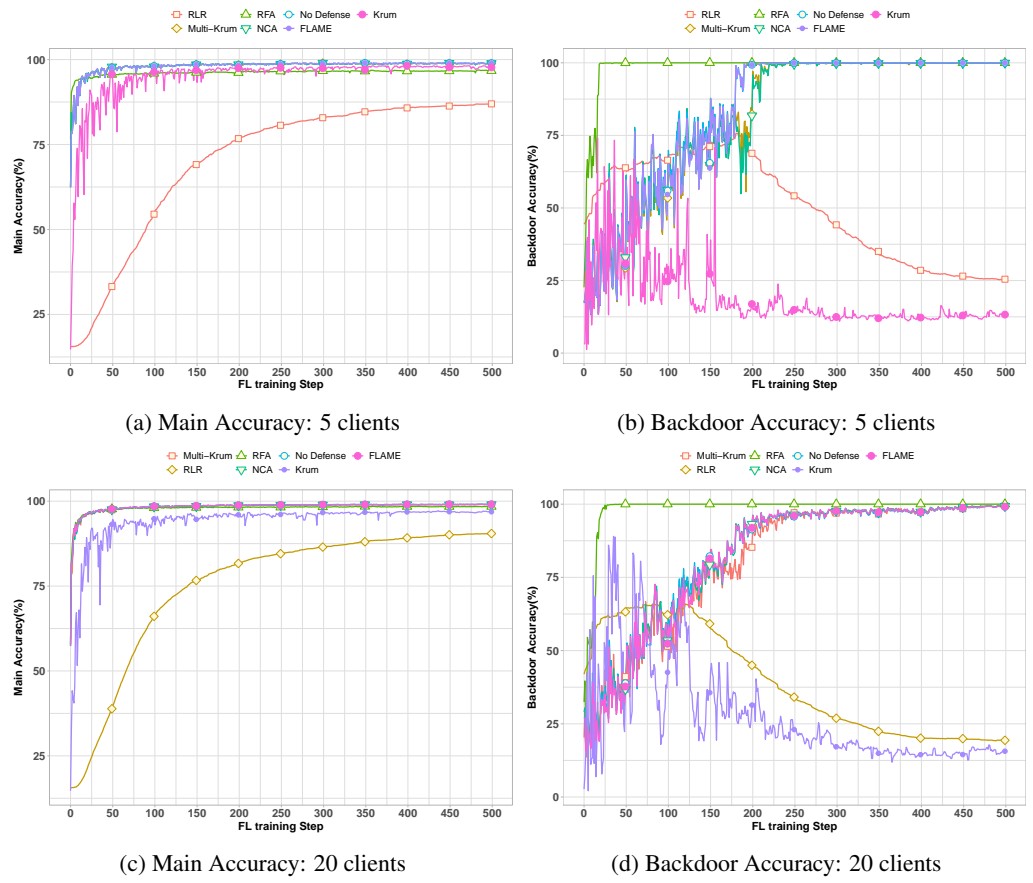

(a) Main Accuracy: 5 clients

(b) Backdoor Accuracy: 5 clients

(c) Main Accuracy: 20 clients

(d) Backdoor Accuracy: 20 clients

Figure 6: The performance of IBA (stand-alone) under different numbers of participating clients $K$.

### B.3 IBA in the scenario of fixed-pool attacks

In this experiment, we investigate the performance of IBA under fixed-pool backdoor attacks, in which $\mathcal{A}$ compromises a fraction of clients in FL, i.e., $\epsilon$. In this scenario, each compromised client has its local generator, and the global generator is calculated using the average operator of the parameters of the local generators. This concept resembles the FedAvg aggregator. The experiment in CIFAR-10 demonstrated a clear trade-off between MA and BA as depicted in Figure 8. As the number of compromising clients increased, BA also increased while MA gradually decreased. We can see the impact of the compromising ratio in Figure 7, in which the higher $\epsilon$ can significantly boost the backdoor effect with both experimented datasets. Notably, even with small fraction of compromised clients, i.e., 2.5%, IBA can achieve at least more than 85% of BA.

To maintain a balance between MA and BA, a compromising ratio of $\epsilon = 10\%$ was chosen for further analysis. The results presented in Figure 9 revealed that there were no significant differences in the MA and BA values across the three datasets (MNIST, CIFAR-10, and Tiny ImageNet) with a compromising ratio of $\epsilon = 10\%$. Significantly, the participation of multiple malicious clients leads to faster convergence of the global attack model. This observation highlights the consistency of the IBA approach in maintaining comparable levels of MA and BA across different datasets and attack scenarios.

### B.4 Durability Evaluation

We evaluate the backdoor effect of the proposed IBA as well as a SOTA backdoor attack, namely, DBA [18]. The scenario is that we allow the malicious client(s) to participate in the training continually over a set number of rounds. We conducted the comparison in the same FL environment simulation and training parameters such as batch size and number of local training epochs $E$ are

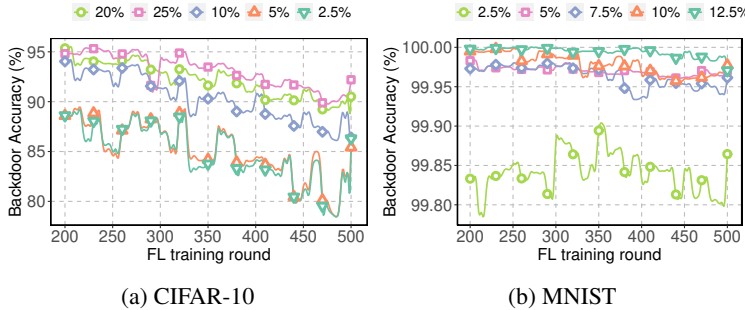

Figure 7: Backdoor accuracy's variation by round for different $\epsilon$.

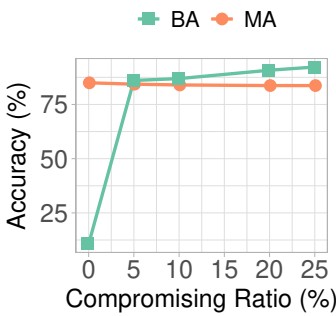

Figure 8: IBA under fixed-pool attacks with different compromising ratios.

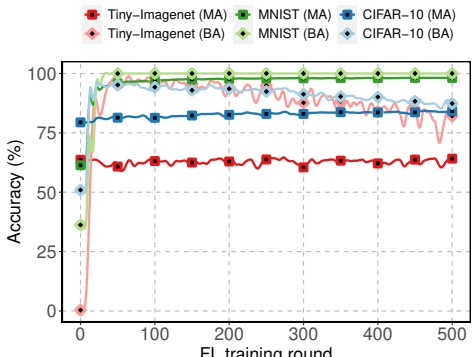

Figure 9: By-round accuracy of fixed-pool IBA with compromising ratio $\epsilon = 10\%$.

kept the same. With the DBA attack, four malicious clients, each with its own local trigger, will participate in the training for every 10 rounds until round 400/ 200 for the CIFAR-10/ MNIST datasets, respectively. In terms of IBA, we allowed a single malicious client to participate in the training with the same frequency until round 400/ 200 for CIFAR-10/ MNIST datasets. The malicious client(s) are then completely eliminated from the training process.

Table 5: Performance of DBA and IBA after the malicious client(s) are removed from the training.

| Dataset | Method | Stopping round | | 50 rounds later | | 100 rounds later | | 150 rounds later | | 200 rounds later | | 250 rounds later | |
|---|---|---|---|---|---|---|---|---|---|---|---|---|---|
| | | BA | (%) | BA | % | BA | (%) | BA | (%) | BA | (%) | BA | (%) |
| MNIST | IBA | 99.94 | 100.00 | 99.84 | 99.90 | 99.52 | 99.57 | 99.46 | 99.52 | 99.18 | 99.24 | 99.08 | 99.14 |
| | DBA | 99.70 | 100.00 | 98.44 | 98.74 | 91.44 | 91.71 | 79.88 | 80.12 | 65.04 | 65.24 | 38.64 | 38.76 |
| CIFAR-10 | IBA | 87.51 | 100.00 | 86.22 | 98.53 | 85.32 | 97.50 | 84.89 | 97.00 | 83.19 | 95.06 | 83.16 | 95.03 |
| | DBA | 69.63 | 100.00 | 65.55 | 94.14 | 63.59 | 91.32 | 55.39 | 79.55 | 55.07 | 71.91 | 42.55 | 61.11 |

As demonstrated in Table 5, IBA delivers superior backdoor efficiency and durability within the same attacking duration. After round 400, the BA of DBA drops quickly compared to IBA. We can see the most significant difference towards the end of the comparison duration, i.e., at round 700, the BAs of IBA and DBA are $\sim 85\%$ and $\sim 45\%$, respectively. After 250 rounds of elimination, the BA of DBA retains 38.76% of its peak value, but the comparable value of IBA is 99.14%. Furthermore, while the BA can preserve more than 60% of its peak value with the CIFAR-10 dataset, the highest accuracy it can attain is significantly lower than IBA, i.e., less than 20%, which is consistent with the prior conclusion of the original when the scaling factor is 1.

To this end, the IBA outperforms the DBA in terms of durability and efficiency, demonstrating the more superior impact of learning optimal trigger generation over the conventional homogenous trigger in prior studies.

## C   Ablation Study

### C.1   Extended Analysis of IBA's Longevity

We conducted additional experiments to assess the durability of our proposed method, IBA, in comparison to other techniques, namely 3DFed [9] and Edge-case [17]. Notably, our approach consistently outperformed the alternatives in terms of durability, as illustrated in Figure 10. For a fair assessment of these methods in scenarios characterized by a low participation rate (specifically, an attack frequency of 10 rounds), we employed a fixed-frequency attack. Our observations revealed that the Backdoor Accuracy of IBA exhibited remarkable stability, with negligible fluctuations persisting until the round of 1300. This implies that the backdoor effect can be maintained unchanged for more than 1000 rounds, highlighting the robustness and effectiveness of our approach.

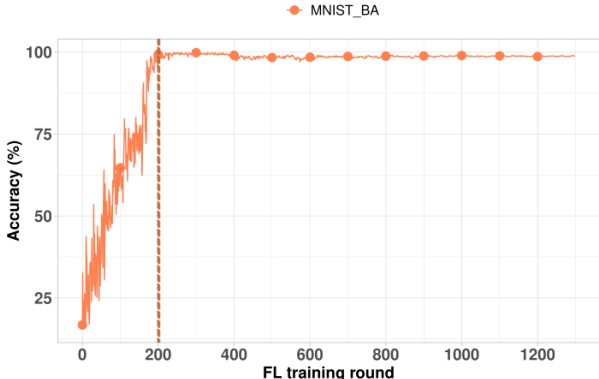

Figure 10: Durability of IBA after backdoor removal with MNIST dataset

### C.2   Neurotoxin Comparision

Since Neurotoxin [19] is a SOTA model poisoning technique in FL, we conducted additional experiments to investigate the efficiency it when it is used as an add-on technique for existing FL backdoor attacks. Specifically, we consider two combinations: (i) centralized backdoor attack (CBA) with Neurotoxin and (ii) distributed backdoor attack (DBA) with Neurotoxin. In the experiment with CBA, the adversary uses a patched trigger to create the backdoor attack and participate with a frequency of 10 rounds. This backdoor trigger is broadly employed in prior works [19, 1]. The second experiment is conducted with four malicious clients participating in every 10 rounds and each client has its own local backdoor trigger which partially constructs the global trigger. Then, each malicious client uses Neurotoxin as a model poisoning technique during the local training process. These experiments are studied under the same setting as our method (1 attacker participating in every 10 rounds).

As observed from Figure 11b, the combination of DBA and Neurotoxin does not have a good effect on extending the backdoor effect. In specific, the BA reduces gradually after round 600, at which point the adversaries leave the training. Moreover, this combination even brings down the main task's accuracy. On the other hand, as shown in Figure 11a, our proposed model poisoning method brings better durability compared to the original Neurotoxin, i.e., after round 400 at which the adversary leaves the training, the IBA+Neurotoxin's BA drops more quickly.

### C.3   Longevity of 3DFed

To shed more light on this aspect, we conducted a comparative analysis with another SOTA backdoor attack in FL,i.e., 3DFed [9]. 3DFed is designed as a multi-layered evasion structure that obtains feedback (i.e., gradient updates) from the global model on the attack to find the redundant neurons to further leverage norm-clipping and masking techniques to enhance the robustness of the attack. Our investigation concentrated on assessing the method's resilience post-attack over 200 rounds. As observed from Figure 12, 3DFed revealed that the model's proficiency in learning backdoor tasks

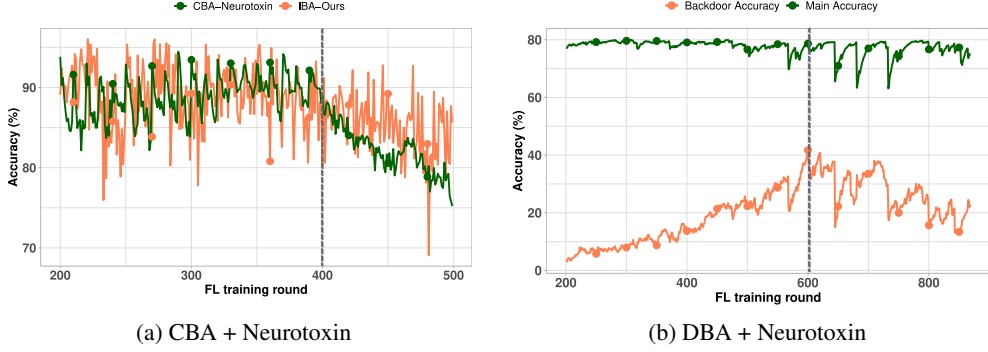

(a) CBA + Neurotoxin

(b) DBA + Neurotoxin

Figure 11: Further comparison of IBA and Neurotoxin

eroded significantly. After 50 epochs and reaching round 250, accuracy plummeted from 85.98% to 9.69% on MNIST, and from 75.92% to 6.06% on CIFAR-10. Since the key to 3DFed's success is dependent on the feedback from the global model to adapt their attack parameters automatically, when the adversarial clients are removed from the training, the backdoor effect will be diluted. In stark contrast, our IBA method demonstrated remarkable persistence. Even at the round of 450, accuracy remained below 1% for MNIST and below 5% for CIFAR-10. Notably, when compared to 3DFed and DBA, IBA consistently outperformed them, as they exhibited an incremental accuracy decline beyond the round of 450. This underscores IBA's exceptional efficacy in upholding model robustness amidst adversarial conditions.

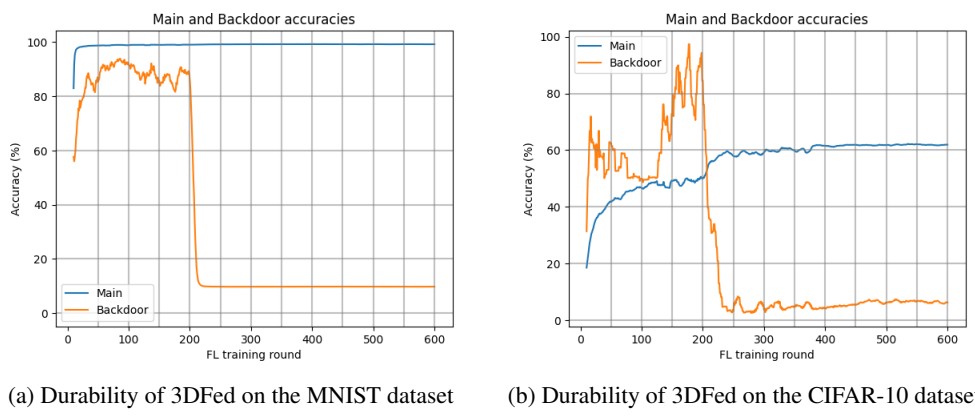

(a) Durability of 3DFed on the MNIST dataset

(b) Durability of 3DFed on the CIFAR-10 dataset

Figure 12: Durability of 3DFed Backdoor Attack

### C.4 Edge-case Backdoor Attacks

Even though we were initially interested in the limitations of existing artificial backdoor attacks (i.e., trigger-based backdoor attacks), we did one more study on the durability of trigger-less backdoor attacks (i.e., the edge-case at [17]) to show that our proposed attack IBA has a longer lifespan. One can see from Figure 13 the gradual degradation of the edge-case backdoor attack when the adversary is removed from round 400.

## D   Limitations

To the best of our knowledge, IBA is the first work that investigates realistic assumptions, imperceptibility, and durability in FL backdoor attacks utilizing optimum trigger-generating function learning. However, the adversary must be able to insert digitally generated triggers into the images before providing them to the classifier. Additionally, although the performance of IBA is only proved with image classification tasks, the idea of learning a trigger-generating function can be expanded to other

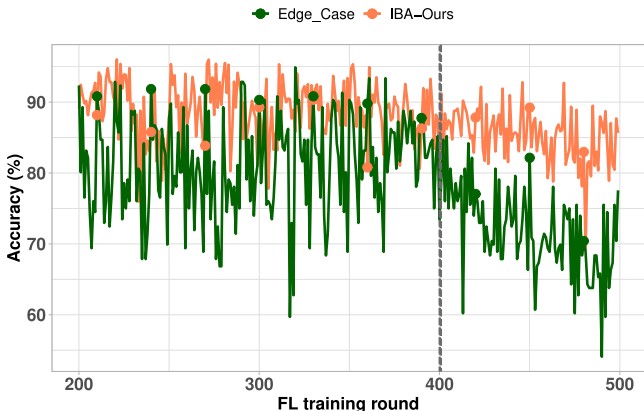

Figure 13: Durability of Edge-case Backdoor Attacks with CIFAR-10 dataset

domains such as NLP and IoT. Therefore, extending the learnable trigger situation into physical attacks and its applicability in other domains is an intriguing future option. Such assessments can help determine if IBA is only a theoretical phenomenon or a real-world threat.

Besides, the distributed nature of FL has yet to be fully exploited in IBA because we concentrate on dealing with the attack of a single malicious client. A better cooperatively attacking approach may be researched further to produce even more impressive results in backdooring the model.

Because this is the first effort in this area, there is no current defense that addresses the scenario of IBA, and our studies indicate that the existing common defensive approaches are ineffective against IBA. On the other hand, these experiments do not reasonably or effectively evaluate the performance of our approach. To this end, we encourage future research on developing more powerful defenses to combat our stealthy backdoor attack with significantly enhanced adversarial capabilities.

## E Societal Impacts

Our work is likely to increase the awareness and understanding of such vulnerability when training neural networks in the FL setting. The proposed attack may bring harm to existing FL applications if it is not appropriately used or stronger mitigation methods are not available. We believe our work is an important step towards understanding the full capability of backdoor attacks in the FL environment. We hope that this finding will, in turn, facilitate further developments of secure FL models and more powerful defensive solutions.