# OpenReview forum: "IBA: Towards Irreversible Backdoor Attacks in Federated Learning"
_NeurIPS.cc/2023/Conference — NeurIPS 2023 poster_

### Official Review · Reviewer_Avr8 · 2023-06-18

**Soundness:** 2 fair
**Presentation:** 2 fair
**Contribution:** 1 poor
**Rating:** 4
**Confidence:** 5

**Summary:**

The paper introduces a two-phased backdoor injection framework, called IBA, for Federated Learning systems. IBA incorporates an adaptive trigger generation mechanism along with a gradual implantation process to insert stealthy backdoors into the global model. IBA enhances the efficiency and durability of the attack through selective poisoning of specific model parameters. Through evaluation using multiple datasets, IBA demonstrates high success rates and outperforms existing backdoor injection methods even in the presence of several defense techniques.

**Strengths:**

The paper is well-written and easy to follow. It addresses a crucial problem in federated learning.

**Weaknesses:**

The paper has several limitations which are discussed below:

The primary contribution of the paper is to use adversarial examples for generating adaptive backdoor triggers and selective parameter poisoning for injecting durable backdoors. However, a quick search resulted in the following two papers:

[1] M. Alam et al., "PerDoor: Persistent Non-Uniform Backdoors in Federated Learning using Adversarial Perturbations", arXiv 2022.

[2] Z. Zhang et al., "Neurotoxin: Durable Backdoors in Federated Learning", ICML 2022.

It is evident from these papers that [1] uses adversarial examples as backdoor triggers, and [2] uses selective parameter poisoning for durable backdoors. On top of that, [2] performs similar operations as the authors do to increase the durability of backdoors. The authors should elaborate on the distinctiveness and novelty of their approach compared to these preceding works.

IBA utilizes adversarial examples as backdoor triggers, which are known for their input specificity. The paper could benefit from a more comprehensive explanation of how IBA addresses this specificity—questions such as whether triggers are unique to each input or shared across a class. Also, it needs to be elaborated more on whether the trigger generation function must be trained at each training round. If so, what is the computational overhead for that?

IBA is not evaluated against important state-of-the-art defenses like FLAME[3], SparseFed[4], etc.

IBA should be compared against other state-of-the-art attacks like 3DFed[5].

IBA should be evaluated using more practical benchmark federated learning datasets in the LEAF project[6].

The evaluation of IBA considering the fixed-pool case should be included as the main content of the paper.

The reported accuracy of 98.09% on the CIFAR-10 dataset and the subsequent 14% drop in accuracy in Table 1 raise skepticism, especially in comparison to the marginal drop observed for T-Imagenet. The authors should provide clarification and insights into this discrepancy. In addition, there appears to be a mismatch between the content of Table 1 and its explanation in Section 4.2 (1). The authors need to ensure consistency and clarity in the representation and discussion of results.

The authors should provide a detailed explanation for specific results, such as why the combination of IBA and PGD yields an accuracy of 18.74% for MNIST and RLR in Table 2 and other similar cases. With such low accuracy, why is IBA deemed to bypass every defense mechanism?

The authors mention that "This trigger is expected to bypass human and machine inspections" but fail to discuss the metrics or criteria used for this assessment.

Figure 1 lacks an adequate explanatory description. A more thorough discussion will enhance the clarity and comprehensibility of the figure.

To provide more insights, it would be beneficial to depict results on more complex color images in Figure 4 instead of using a monochromatic MNIST dataset.

The authors should discuss the significance of using FedProx and FedNova, and clearly describe how these differ from FedAvg. This would help in understanding the rationale behind their inclusion.

What does the anomaly index represent in Figure 6?

NeuralCleanse is mentioned as a defense mechanism against backdoor attacks in FL, but it was not originally designed for the FL framework. Clarifying its relevance and applicability in this context would be beneficial.

[3] T. Nguyen et al., "FLAME: Taming Backdoors in Federated Learning", Usenix Security 2022.

[4] A Panda et al., "SparseFed: Mitigating Model Poisoning Attacks in Federated Learning with Sparsification", AISTATS 2022.

[5] H. Li et al., "3DFed: Adaptive and Extensible Framework for Covert Backdoor Attack in Federated Learning", IEEE S&P 2023.

[6] https://leaf.cmu.edu/

**Questions:**

Please address the issues discussed in Weaknesses.

**Limitations:**

The authors have discussed the limitations and potential negative societal impacts of the paper. However, the paper has several other limitations, as discussed in the review.

---

> ### Author Rebuttal · Authors · 2023-08-10
>
> We appreciate the valuable comments. Additional experiments are at [**EXP**](https://files.fm/f/r67m5e7a3).
>
> 1. **Compare with PerDoor and Neurotoxin**
>
> PerDoor uses the Basic Iterative Method to generate the trigger and relies on gradients of the loss w.r.t input. Alternatively, we learn a generative model to generate the trigger that blends well with the underlying data distribution. Moreover, the collaborative clients learn their own generative models but aggregate them into one shared model in the next rounds. PerDoor lacks this sharing mechanism, which makes the attack more effective and durable.
>
> While the idea of gradient masking is explored in~\cite[36, 37], our usage of historical information to determine the gradient mask is novel. Our use of a historical gradient mask helps reduce the bad effect of the fluctuation of the global model on finding the optimal gradient mask to poison, especially for non-iid clients with skewed data distributions (may temporarily distort the global behavior). The additional experiments in EXP show the superior performance of IBA's poisoning compared to neurotoxin.
>
> 2. **Computation overhead**
>
> As presented, $T(x) = x+G(x)$, $\|G(x)\|$, $x$, for each input $x$, and G learns to generate an input-aware trigger. However, G is updated after the local model’s training is completed, which does not cause a delay in local model submission. In other words, the computational overhead can be easily alleviated in a different thread or outside of FL training.
>
> 3. **Evaluation against FLAME/SparseFed**
>
> We performed additional experiments with the FLAME in EXP, and show that even stand-alone IBA can get around it. IBA achieves 99% BA on MNIST for both cases of K=5 and K=20. Our considered threat model is different from that of  FedSparse, where the server does not send the identical global model to each device.
>
> 4. **Comparison with 3DFed**
>
> We tested IBA against the SOTA 3DFed [5]. We evaluated the method's durability post-attack over 200 rounds (see Figure 3 of EXP). Notably, after 50 epochs and reaching round 250, 3DFed model's ability to learn backdoor tasks deteriorated significantly. Accuracy dropped from 85.98% to 9.69% on MNIST and from 75.92% to 6.06% on CIFAR-10. However, IBA still has superior resilience. Even at round 450, accuracy remained below 1% for MNIST and 5% for CIFAR-10. IBA outperformed both 3DFed and DBA, which exhibited a gradual accuracy decline beyond round 450.
>
> 5. **Evaluation with LEAF**
>
> We employed GitHub's "Attack of the Tails," as the foundation for implementing and evaluating IBA, to ensure a reliable evaluation within an established framework. However, assessing IBA in more FL datasets and benchmark is an interesting future direction of our work, as suggested.
>
> 6. **Evaluation of IBA considering the fixed-pool case**
>
> We will revise this part in the later version, as suggested.
>
> 7. **Accuracy drop on CIFAR**
>
> We confirm that the MA of CIFAR-10 (baseline case w/o. attack) should be 84.71% (a typo in the current version). We will revise it accordingly.
>
> 8. **Explanation for specific results**
>
> While Krum and RLR defend the proposed attack to some extent, i.e., we observe the drop in the BAs of IBA, we argue that Krum and RLR tend to reject previously unseen information from both adversary and honest clients, which results in low MAs (a drop of 3–10%). Specifically, the proposed IBA causes RLR to compensate for the backdoor defense’s effectiveness with the main accuracy drop in the MNIST dataset by unnecessarily reversing the learning rate of specific dimensions, i.e., the BAs are around 90% (less than other defenses 10%). Therefore, these two defenses raise concerns about their practicality.
>
> 9. **Metrics for assessing bypassing human inspections**
>
> We follow WaNet to perform a similar human evaluation experiment: we tell the testers (a cohort of 50) the mechanism of the attack (e.g., how the trigger is created), present to a tester a clean image and its corresponding backdoor image (with the trigger) without revealing the images’ identities, and ask them to identify the backdoor image. The result shows that the testers cannot distinguish which images are clean and backdoor, suggesting that our trigger is imperceptible against human inspection.
>
> 10. **Explanation for Fig 1**
>
> We added one paragraph to explain Fig 1 in Sec 3: Irreversible Backdoor Attacks (IBA) in FL. In the trigger-generating stage, IBA trains a generative trigger model. In the second stage, the attacker joins the FL process and trains a backdoored model with the triggers. The loss function of IBA is the combination of the loss function of the backdoored data and the loss function of the benign data. The objective of the attacker is to maximize the accuracy of the backdoored model on the training set over the threshold (i.e., 0.85).
>
> 11. **Results on color images**
> We included include the GradCAM analysis for CIFAR-10 (in EXP). We can observe that the backdoored images have negligible deviations from the Grad-CAM behaviors on the clean images. On the other hand, backdoored images using patched triggers generate a considerable difference in visualization heat maps (DBA [31]).
>
> 12. **using FedProx and FedNova**
>
> We will discuss FedProx and FedNova for future extensions of our work in the later version. Evaluating IBA against FedProx and FedNova is an interesting extension of our work.
>
> 13, 14. **Anomaly index in Figure 6 and NeuralCleanse in FL**
>
> NC reverse-engineers the assumed patch-based trigger and uses the Anomaly Index to identify an anomalous trigger that is smaller than the rest. If the Anomaly Index < 2 for a class, NC tags the model as backdoor with this class as the target label. NC's Index on IBA is less than 2 or the benign model in T-Imagenet. This means that NC is not effective against IBA.
>
> [36] Zhou et al. "Deep model poisoning attack on federated learning."
>
> [37] Zhang et al. "Neurotoxin: Durable backdoors in federated learning."

---

> ### Comment · Reviewer_Avr8 · 2023-08-21
>
> I want to thank the authors for their efforts in responding to these queries. I respect the hard work put into this paper and trust that these suggestions will only enhance its quality. I am satisfied with most of the responses and am increasing my score.

---

> > ### Author Response · Authors · 2023-08-21
> >
> > We are glad to answer all of the questions. Thank you very much for your insightful comments and kind help. Our work proposes a novel backdoor attack on federated learning that achieves multiple important objectives, including effectiveness, stealthiness again human/ machine inspections, and durability with intensive experiments. This work takes an important step towards understanding the extensive risks of backdoor attacks in FL, urging practitioners to investigate more effective backdoor mitigation methods in the FL domain.

---

### Official Review · Reviewer_yF1b · 2023-07-04

**Soundness:** 2 fair
**Presentation:** 3 good
**Contribution:** 2 fair
**Rating:** 4
**Confidence:** 4

**Summary:**

This paper studies backdoor attacks in federated learning setting. They propose a new backdoor attack method that is based on sample-specific trigger, optimized with constraints on weight norm and weight dimension, to achieve more stealthy, harder to detect and more durable. U-Net is trained to generate sample-specific trigger and the stealthiness of the trigger is controlled by $\epsilon$, which also controls the balance between attack success and trigger stealthiness. To make the attack harder to detect and more durable, the gradients are projected to the ball around the global model weights and optimization is constrained to dimensions that are historically infrequently updated. Experiments on CIFAR10, MNIST and TinyImageNet show that the proposed method is effective against multiple state-of-the-art defense methods.

**Strengths:**

- The paper is well organized and the ideas are clearly presented.

- Experiments show that the method can defend against multiple state-of-the-art defense methods.

**Weaknesses:**

- Even though the proposed method is compared with DBA to show its durability, the paper lacks comparison with other state-of-the-art attack methods. Experiments comparing multiple attack methods against state-of-the-art defense methods on different types of datasets should be conducted to show the effectiveness of the method.

- As for the poisoning dimension restriction, the idea of restricting updates to dimensions that are historically seldomly updated is similar to that of Neurotoxin. Why compare with DBA which does nothing to promote the durability of the attack? Why not compare with Neurotoxin?

-  Generally speaking, the data distribution of clients may affect the performance of attack and defense. Dir(0.5) and Dir(0.01) are used for MNIST, CIFAR10/TinyImagenet. Experiments studying the effect of data distribution could be helpful understand the method better.

- Based on Figure 2 and table 2, it seems that the method does not do a good job against Krum and RLR. The attack accuracy of the method keeps dropping when tested against the two defense methods.

**Questions:**

See questions and concerns in the previous part.

**Limitations:**

The main limitation is the lack of experiments to support the claim. See details in weakness part.

---

> ### Author Rebuttal · Authors · 2023-08-10
>
> We appreciate the comments and suggestions from the reviewer. The following is our response to the raised concerns.
>
>
> 1. **Comparison with other state-of-the-art attack methods**
>
>
> As mentioned, our objective is to design a backdoor attack with efficiency, stealthiness, and durability under more practical assumptions. Therefore, regarding evaluating efficiency and stealthiness, we refer to the experimental design in prior works [1, 26, 31] to understand the performance of IBA under different attack schemes (fixed-frequency vs. fixed-pool) and mainstream backdoor defenses. With respect to durability, we selected DBA (a SOTA backdoor attack in FL) and configured it such that two attacks are under the same participation frequency, and we observed the behavior of backdoor accuracy after the adversary is removed. Based on the two main designs above, we want to draw the conclusion that IBA can achieve equivalent results to other attacks such as Edge-case [26], DBA [31] under practical assumptions but is stealthier in terms of visual representation and more durable. To further demonstrate the superior durability of the proposed IBA, we compared it with 3DFed [34], a new SOTA backdoor attack in FL and proved that IBA can bring much better durability and performance.
>
>
> 2. **Comparison with Neurotoxin**
>
>
> We conducted additional experiments to compare the durability of the proposed attacks with Neurotoxin.
> - IBA + proposed model poisoning (Ours) (1)
> - DBA + Neurotoxin (2)
> - Centralized backdoor attacks + Neurotoxin (3)
>
>
> As observed, the combination of DBA and Neurotoxin (2) does not have a good effect on extending the backdoor effect. In specific, the BA reduces gradually after round 600, at which point the adversaries leave the training. Moreover, this combination even brings down the main task's accuracy.
>
>
> We consider the standard centralized backdoor attack [31, 1] combined with Neurotoxin (3) under the same setting as our method (1 attacker participating in each 10 rounds). In the experiment with Neurotoxin, the adversary uses a patched trigger to create the backdoor attack.
> From the result, our proposed model poisoning method brings better durability compared to the original Neurotoxin, i.e., after round 400 at which the adversary leaves the training, the IBA+Neurotoxin’s BA drops more quickly.
>
> For comparison of IBA and Neurotoxin over FL training rounds, please refer to Figure 3 in `Extended Durability Evaluation` section in this [**pdf**](https://files.fm/f/r67m5e7a3) file.
>
>
> 3. **Effect of data distribution**
>
>
> We consider different value of alpha (0.2, 0.5, 0.7) in the Dirichlet distribution to simulate from non-i.i.d to i.i.d distributions for the image datasets. When evaluated under 2 different datasets, the result shows that IBA-BA is stable under various distributions, which exhibits the practicability and robustness of IBA when attacking standard FL. The table below shows the results of IBA-BA under different alpha values in the Dirichlet distribution.
>
>
> **Main Accuracy** (MA) of IBA under different alpha values in the Dirichlet distribution.
>
>
> | Dataset | Alpha = 0.2 | Alpha = 0.5 | Alpha = 0.7 |
> | --- | --- | --- | --- |
> | MNIST | 99.01% | 99.98% | 98.99% |
> | CIFAR-10 | 82.73% | 84.28% | 84.53% |
>
>
> **Backdoor Accuracy** (BA) of IBA under different alpha values in the Dirichlet distribution.
>
>
> | Dataset | Alpha = 0.2 | Alpha = 0.5 | Alpha = 0.7 |
> | --- | --- | --- | --- |
> | MNIST | 98.89% | 98.03% | 98.81% |
> | CIFAR-10 | 83.26% | 85.63% | 88.93% |
>
>
> 4. **Effectiveness against Krum and RLR**
>
>
> While Krum and RLR defend the proposed attack to some extent, i.e., we observe the drop in the BAs of IBA, we argue that Krum and RLR tend to reject previously unseen information from both adversary and honest clients, which results in low MAs (a drop of 3-5%). Therefore, these two defenses raise a concern for their aplicability under the proposed attacks.

---

### Official Review · Reviewer_S7Me · 2023-07-06

**Soundness:** 3 good
**Presentation:** 2 fair
**Contribution:** 2 fair
**Rating:** 6
**Confidence:** 4

**Summary:**

The authors propose a backdoor attack framework (IBA) in Federated Learning for trigger based backdoors that jointly learns a generative model for stealthy visual triggers while also planting the backdoor in the global model. They evaluate their attack on MINST, CIFAR-10, and TinyImageNet and show that it bypasses several known defenses such as KRUM, NC, RFA etc.

**Strengths:**

1. The experimental setup is quite thorough, considering several datasets, settings, and defenses.
2. The attack appears to bypass all the considered defenses with good effectiveness and stealth.
3. The work acknowledges past work well with a good description of existing research.

**Weaknesses:**

1. While the attack is titled "Irreversible Backdoor attack", the longevity of the attack is not well studied. Figure 5 is only for the MNIST dataset and most other experiments consider fixed-frequency attack. I would like to see a better analysis of how long the attack takes to be removed by clean training.
2. I would like to see a comparison to even trigger-less backdoor attacks since the PGD model replacement attack has already been well studied in Bagdasaryan et al. and Wang et al. I would expect this to be much stronger, but it would act as a good baseline. While there are several ablations, I would like to see comparison to more existing baselines. Perhaps even BadNets [Gu et al.].

**Questions:**

1. In (2), the adversarial perturbation is bounded in the $\lVert \cdot \rVert_{\infty}$ norm. Can this be translated to the 2-norm? Naively, speaking the same attack budget will get scaled down by $\sqrt{d}$, which will reduce the effectiveness of the attack. Have you tried this?
2. It is quite interesting that the optimal choice of scaling factors is $\alpha = \beta = 0.5$. Are the scales of the two losses very similar?
3. How is $\lambda_{\xi}$ chosen?
4. Please exwhat plain acronyms in the table captions. Such as MA, BA.
5. Please explain bold entries in the tables mean. It can be inferred, but would be nice not to have it.
6. Section 4.2 is quite hard to understand especially since accuracies are listed as "ranging" across datasets. This is quite unusual and doesn't really make sense.
7. What is MR?
8 . It is hard to evaluate the performance of IBA from Table 2 without comparing against other attacks.
9. It seems like for some defenses (like KRUM), IBA + PGD outperforms IBA + PGD + MR. Whereas in RLR, it gets worse. Can you explain this behavior? Is one supposed to choose the attack based on a defense?
10. In Figure 3, is the attack at round 0?
11. Can you specify the choice of $\varepsilon$ for the PGD attack?
12. Some citations are duplicated (McMahan et al., Wang et al)

**Limitations:**

I would strongly recommend the authors include a section listing the limitations of their work (comparison with more baselines, longevity of attack, need for whitebox access etc)

---

> ### Author Rebuttal · Authors · 2023-08-10
>
> Thank you for the valuable comments. Additional experiments are at [**EXP**](https://files.fm/f/jh4hy9pg3).
>
> 1. **analysis of longevity**
>
> As presented in Fig 5 (main), Fig 9 (sup.), and Tab 4 (sup.), our method can achieve extended longevity compared to DBA for both MNIST and CIFAR-10. We also conducted additional experiments to compare the durability of IBA, 3DFed [1], and Edge-case [2626], and ours still outperformed the others w.r.t durability (results in Fig 2 of EXP). We leverage the fixed-frequency attack for a fair evaluation of the methods under a low participation rate (i.e., attack frequency is 10 rounds).
>
> 2. **Compare to more baselines**
>
> We provided additional study on the durability of trigger-less backdoor attacks (i.e., the edge-case at Wang et al.) to show that our proposed attack IBA has a longer lifespan (Fig 5 in the provided file).
>
> Since patched triggers used in DBA are relevant to the methodology employed in BadNets, we selected one more SOTA, i.e., 3DFed, to compare with ours in terms of durability. IBA outperformed both 3DFed and DBA, which exhibited a gradual accuracy decline beyond round 450. For further durability evaluation over training rounds, please see Fig 4 and 5 in  `Extended Durability Evaluation` of EXP.
>
>
> 3. **Perturbation using $\|.\|_{\infty}$ Norm**
>
> While considering various norms for bounding the generator noise, such as the $L_2$ norm, it's generally inadvisable for backdoor attacks. This infinity norm guarantees a widespread distribution of the generated trigger across the input image (encompassing all pixels in the trigger); in contrast, the $L_2$ norm can result in localized artifacts within the image (with only select pixels forming the trigger). In simpler terms, employing the $L_2$ norm can make the backdoor attack more susceptible to detection by trigger-synthesis defenses like Neural Cleanse. Thus, the $\|.\|_{\infty}$ norm stands as a preferred choice. We will enhance the clarity of this aspect.
>
> 4. **Optimal choice of scaling factors**
>
> As discussed in Sec 3.2 (lines 165-169), empirically, if $\alpha$ is significantly higher than $\beta$, the classifier's performance on clean data rapidly converges to the optimal performance of the vanilla classifier; on the other hand, if $\beta$ is significantly higher than $\alpha$, the classifier's performance on backdoor data quickly reaches the optimal value. Thus, to balance the learning processes of both the main and backdoor tasks, we set $\alpha = \beta = 0.5$ in our experiments.
>
> 5. **How is $\lambda_{\xi}$ chosen?**
>
> $\lambda_{\xi}$ is the decay rate appearing in the formula (4) that controls how fast the reduction by round is. The larger $\lambda_{\xi}$ is, the faster the backdoor images become stealthy and the harder it is for the generative model to learn to perform well (may overfit with the local data). Therefore, our empirical results suggest to fix the value of $\lambda_{\xi}$ to be 0.001 to balance out the two factors mentioned above. Our suggestion is that $(1 - \lambda_{\xi})$ should be set in the range $[1\times (1-\gamma); 3\times (1-\gamma)]$, where $\gamma$ is the decay factor of learning rate, i.e., in our inherited setup, $\gamma$ is set to be 0.998.
>
> 6. **Acronyms in the table captions**
>
> As provided in Sec 4.2, MA and BA stand for Main Accuracy and Backdoor Accuracy, respectively. The bolded entries in the table indicate the best MA/BA achieved by each method for a given dataset. MR stands for Model Replacement [1]. We will make the adjustments accordingly.
>
> 7. **Unusual writing (`ranging` word)**
>
> Here, we aim to show that the MAs of the poisoned models are similar to those of the corresponding clean models while their BAs are much higher under our attack.
>
> 8. **evaluate the IBA performance, Tab 2, with other attacks**
>
> We focus on providing a comprehensive empirical analysis of the proposed attack, including the properties of the designed trigger function, the attack’s stealthiness and durability,  (main concerns of related federated backdoor-research works). Therefore, regarding evaluating efficiency and stealthiness, we refer to the experimental design in prior works [31, 26, 1] to understand the performance of IBA under different attack schemes (fixed-frequency vs. fixed-pool) and mainstream backdoor defenses. The experiment shows that IBA can obtain equivalent BAs as other backdoor attacks, i.e., >90% for 3 datasets, while ensuring its stealthiness in terms of visualization and defense bypassing and extended durability.
>
> 9. **Behavior of IBA+PGD/ IBA+PGD+MR**
>
> In most cases, the combination of IBA + PGD/ IBA + PGD + MR helps improve the backdoor accuracy from 10% to 20%. However, Krum and RLR are the most challenging to circumvent. For Krum, MR seems not to work well under the MNIST dataset since the poisoned model is scaled before being submitted, and Krum chooses the model with the smallest Euclidean distance to its neighbors as the new global model.
>
> For RLR, the behavior of attacks is difficult to predict. At some suspicious dimensions of the whole model parameter, the learning rate will be reversed. If the poisoned model is scaled (by MR) and then reversed, the global model will move farther from the optimal point of the backdoor task.
>
> Nevertheless, IBA can still be effective against Krum and RLR.
>
> 10. **Attack at round 0 in Fig 3**
>
> Fig 3's attack is at round 0. Here,  with fixed-frequency attacks of 10, the attacker participates in round 0, 10, 20...
>
> 11. **Choice of $\epsilon$ for PGD**
>
> The norm-bound for PGD attack should be selected based on the observation of the $L_2$-norm variation of the global model by round. Following the work [1] and [26], the norm bound is set to be 2 for the CIFAR-10/Tiny-Imagenet dataset and 1.5 for the MNIST dataset. We will add this to the updated version.
>
> 12. **Duplicate citations**
>
> We will revise this section.

---

> > ### Comment · Reviewer_S7Me · 2023-08-12
> > **Response to Authors**
> >
> > I thank the authors for the detailed response. I am satisfied with most of the responses.
> >
> > I had indeed overlooked Table 4 in the sup which studies the durability of IBA. However, I noticed something strange in Table 4. The Main accuracy (which I assume is listed under (%)) seems to continue dropping after the malicious clients are removed from the pool. Can you explain this behavior?

---

> > > ### Author Response · Authors · 2023-08-12
> > > **We sincerely thank the reviewer for the valuable comments and the quick response!**
> > >
> > > We are glad to answer all of the previous comments/questions. Please see our responses to the additional question below.
> > >
> > > **Q: The Main accuracy (which I assume is listed under (%)) seems to continue dropping after the malicious clients are removed from the pool.**
> > >
> > > The column (with heading %) next to BA in Table 4 (supp file) denotes the Relative Backdoor Accuracy w.r.t the Backdoor Accuracy obtained at round 200 when the attackers (malicious clients) are removed from training. In other words, it is the measure of durability or how much the backdoor behavior persists after these malicious clients are removed completely from the training process. For example, for MNIST, at 50 rounds after the malicious clients are removed, IBA (with BA 99.84%) retains 99.90% of the original Backdoor Accuracy (99.94%); at 100 rounds later, 99.57% retention; at 250 rounds later, 99.14% retention... This shows the significant durability of IBA's attack, compared to DBA with only 38.76% retention at 250 rounds post-removal.
> > >
> > > We will revise the supplementary document accordingly to make this discussion clearer.

---

> > > > ### Comment · Reviewer_S7Me · 2023-08-12
> > > > **Thank you for the clarification**
> > > >
> > > > Thank you for the clarification. That makes sense! I am satisfied with the response and am increasing my score.

---

> > > > > ### Author Response · Authors · 2023-08-21
> > > > >
> > > > > Dear Reviewer S7Me,
> > > > >
> > > > > We would like to thank you for your valuable comments, suggestions, and increased score. To summarize, we will include the following changes in the camera-ready version of the paper:
> > > > > - Longevity Analysis in Supplementary Material
> > > > > - Comparative Baselines with trigger-less backdoor attacks, highlighting IBA's extended lifespan in Supplementary Material
> > > > > - Acronyms in the table captions and unusual writing (ranging words) in the main paper
> > > > > - Behavior of IBA+PGD/ IBA+PGD+MR in supplementary material
> > > > > - Duplicate citations in the main paper

---

### Official Review · Reviewer_gShs · 2023-07-07

**Soundness:** 4 excellent
**Presentation:** 4 excellent
**Contribution:** 4 excellent
**Rating:** 7
**Confidence:** 3

**Summary:**

**Key Contribution:** This work proposes a two-staged model poisoning attack which works even with only the participation of a small number of malicious clients. Moreover, the proposed method is effective and robust against existing defenses.
The attack is evaluated on a variety of existing defenses on CIFAR-10, MNIST, and Tiny-Imagenet.


**Strengths:**

**Novelty and Significance**
The paper’s two-stage approach is unique and powerful. Learning a generative model for adversarial noise is additionally alarming as it is difficult to detect and defend against.

**Clarity:**
The paper is well-written and clear - the mathematical formulations are simple and easy to follow, with intuitive results. Moreover, the proposed augmentations to the base attack which help evade existing defenses are methodical and clearly motivated.


**Weaknesses:**

**Limited Evaluation:** The paper could be stronger by presenting more results under different numbers of clients. However, despite this weakness, I am overall satisfied with the quality and comprehensiveness of the evaluation.

**Questions:**

- Have you considered different choices of trigger generation models?

- What was the distribution of data considered across the clients during evaluation? Have you considered evaluating with non-iid data?


**Limitations:**

The authors have adequately discussed limitations and potential social impacts of their work.

---

> ### Author Rebuttal · Authors · 2023-08-10
>
> Thank you for your our time and effort reviewing the paper. We appreciate the review and comments on the paper. We would like to address the concerns as follows.
>
> 1. **Results under different number of clients**
>
>     The standard settings are inherited from other works [1, 31], which are 10 clients. In this work, we consider the case when the number of clients increases and decreases. The performance of IBA is consistent under different number of participating clients (5 and 20 clients).
>
>     For performance of IBA (stand-alone) under different numbers of participating clients K over 500 rounds, please refer to Figure 1
>     from in this [**pdf**](https://files.fm/f/r67m5e7a3) file.
>
> 2. **Choices of trigger generation models**
>
>     In general, the generator function can be modeled as an autoencoder or the more complex U-Net architecture [18]. However, we observe a negligible performance difference between using an autoencoder and U-Net. We conjecture that since learning to generate trigger noises is a much simpler task than learning to generate images, using the simpler autoencoder is sufficient. Furthermore, training and performing inference with the autoencoder have lower computational overhead. For these reasons, we employ the autoencoder in all experiments in our paper. We will add this discussion to a later version.
>
> 3. **Data distribution across the clients during the evaluation**
>
>     FL often presumes non-i.i.d. data distribution across parties. Here, we use a Dirichlet distribution (Minka, 2000) with different hyperparameter $\alpha$ to generate different data distribution following the setups in (Bagdasaryan et al., 2018). Specifically, we follow an established evaluation protocol in previous works [26, 31] and simulate heterogeneous data partitioning by sampling $p_k \sim \text{Dir}_K(0.5)/\text{Dir}_K(0.01)$ for MNIST, CIFAR-10/Tiny-Imagenet and allocating a proportion of each class to participating clients.
>
> [1] Bagdasaryan, E., Veit, A., Hua, Y., Estrin, D., & Shmatikov, V. (2018). How to backdoor federated learning. arXiv preprint arXiv:1807.00459.
>
> [18] Olaf Ronneberger, Philipp Fischer, and Thomas Brox. U-net: Convolutional networks for biomedical image segmentation. In Nassir Navab, Joachim Hornegger, William M. Wells, and Alejandro F. Frangi, editors, Medical Image Computing and Computer-Assisted Intervention – MICCAI 2015, pages 234–241, Cham, 2015. Springer International Publishing.
>
> [26] Hongyi Wang, Kartik Sreenivasan, Shashank Rajput, Harit Vishwakarma, Saurabh Agarwal, Jy-yong Sohn, Kangwook Lee, and Dimitris Papailiopoulos. Attack of the tails: Yes, you really can backdoor federated learning. Advances in Neural Information Processing Systems, 33:16070–16084, 2020.
>
> [31] Chulin Xie, Keli Huang, Pin-Yu Chen, and Bo Li. Dba: Distributed backdoor attacks against federated learning. In International conference on learning representations, 2020.

---

### Author Rebuttal · Authors · 2023-08-10

Thank you the reviewers for the initial comments and questions. We provide the additional experimental evaluations to address the comments of the authors in this [**EXP**](https://files.fm/f/r67m5e7a3) file, including the following experiments:

- Performance when varying the number of Participating Clients
- Extended Longevity/Durability of IBA (including 3DFed).
- Comparisons with Neurotoxin, Edge-case.
- Grad-cam on colored images (CIFAR10).

In the separate responses to each reviewer, we will use this EXP file as a main reference for any additional experiments.

---

### Author Response · Authors · 2023-08-21
**Thank you for your valuable comments and welcome additional questions!**

We want to start by thanking the reviewers once more for providing constructive feedback and raising questions to help us improve the quality of our paper. We have addressed all the questions and suggestions from the reviewers in our rebuttal with new experiments as recommended. We encourage all reviewers to share any further thoughts they might have. We are committed to promptly addressing any additional questions that arise. Finally, we kindly ask for the reviewers to reconsider the evaluation score for our paper. Thank you for your help.

---

### Decision · Program_Chairs · 2023-09-21

**Decision:**

Accept (poster)

**Comment:**

The authors propose a novel, two-staged poisoning attack on federated learning that can bypass current defensive mechanisms. The authors have addressed most of the negative concerns in the rebuttal phase, and they are invited to include the additional results reported in the rebuttal in the final version of the manuscript.